# Frequency-selective control of cortical and subcortical networks by central thalamus

Jia Liu[1†], Hyun Joo Lee[1†], Andrew J Weitz[1,2†], Zhongnan Fang[1,3†], Peter Lin[1], ManKin Choy[1], Robert Fisher[1], Vadim Pinskiy[4], Alexander Tolpygo[4], Partha Mitra[4], Nicholas Schiff[5], Jin Hyung Lee[1,2,3,6]*

[1]Department of Neurology and Neurological Sciences, Stanford University, Stanford, United States; [2]Department of Bioengineering, Stanford University, Stanford, United States; [3]Department of Electrical Engineering, Stanford University, Stanford, United States; [4]Cold Spring Harbor Laboratory, Cold Spring Harbor, United States; [5]Department of Neurology, Weill Cornell Medical College, New York, United States; [6]Department of Neurosurgery, Stanford University, Stanford, United States

*For correspondence: ljinhy@stanford.edu

†These authors contributed equally to this work

Competing interests: The authors declare that no competing interests exist.

**Abstract** Central thalamus plays a critical role in forebrain arousal and organized behavior. However, network-level mechanisms that link its activity to brain state remain enigmatic. Here, we combined optogenetics, fMRI, electrophysiology, and video-EEG monitoring to characterize the central thalamus-driven global brain networks responsible for switching brain state. 40 and 100 Hz stimulations of central thalamus caused widespread activation of forebrain, including frontal cortex, sensorimotor cortex, and striatum, and transitioned the brain to a state of arousal in asleep rats. In contrast, 10 Hz stimulation evoked significantly less activation of forebrain, inhibition of sensory cortex, and behavioral arrest. To investigate possible mechanisms underlying the frequency-dependent cortical inhibition, we performed recordings in zona incerta, where 10, but not 40, Hz stimulation evoked spindle-like oscillations. Importantly, suppressing incertal activity during 10 Hz central thalamus stimulation reduced the evoked cortical inhibition. These findings identify key brain-wide dynamics underlying central thalamus arousal regulation.

## Introduction

The thalamus plays an important role in coordinating global brain signals responsible for cognition and normal waking behavior (*Sherman and Guillery, 1996*; *Llinás et al., 1998*; *Mitchell et al., 2014*). The central thalamus and intralaminar nuclei, in particular, have been postulated to play a critical and unique function in regulating arousal, attention, and goal-directed behavior (*Schiff and Pfaff, 2009*; *Mair et al., 2011*). This idea dates back to the first demonstrations that direct and indirect electrical stimulations of central thalamus control cortical electroencephalography (EEG) dynamics and elicit behavioral transitions between drowsiness/relaxation and wakefulness/attention (*Moruzzi and Magoun, 1949*; *Hunter and Jasper, 1949*; *Fuster, 1958*). Since the initial identification of central thalamus's causal effect on brain state and behavior, significant support for its role in arousal regulation has come from anatomical and histological studies. Steriade and Glenn (*Steriade and Glenn, 1982*) identified a monosynaptic pathway from the mesencephalic reticular formation to the central lateral (CL) and paracentral (PC) nuclei of central thalamus that projects to cerebral cortex and striatum. In addition to this input, the central thalamus receives projections from other arousal systems, including norepinephrinergic innervation from locus coeruleus (*Vogt et al.,*

**eLife digest** The ability to wake up every morning and to fall asleep at night is something that most people take for granted. However, damage to a brain region called the central thalamus can cause a range of consciousness-related disorders, including memory problems, excessive sleeping, and even comas. For example, cell death within the central thalamus has been associated with severely disabled patients following traumatic brain injury.

Previous studies have found that electrically stimulating the neurons in the central thalamus can change whether an animal is drowsy or awake and alert. However, it was not clear whether a single group of neurons in the central thalamus was responsible for switching the brain's state between sleep and wakefulness, or how this would work.

Liu, Lee, Weitz, Fang et al. have now used a technique called optogenetics to stimulate specific neurons in the central thalamus of rats, by using flashes of light. Stimulation was combined with several techniques to monitor the response of other brain regions, including fMRI imaging that shows the activity of the entire brain.

The results showed that rapidly stimulating the neurons in the central thalamus – 40 or 100 times a second – led to widespread brain activity and caused sleeping rats to wake up. In contrast, stimulating the neurons of the central thalamus more slowly – around 10 times a second – suppressed the activity of part of the brain called the sensory cortex and caused rats to enter a seizure-like state of unconsciousness. Further investigation identified a group of inhibitory neurons that the central thalamus interacts with to carry out this suppression.

The results suggest that the central thalamus can either power the brain to an "awake" state or promote a state of unconsciousness, depending on how rapidly its neurons are stimulated. Future work will seek to translate these results to the clinic and investigate how stimulation of the central thalamus can be optimized to reduce cognitive deficits in animal models of traumatic brain injury.

*2008*) and cholinergic innervation from the upper brainstem and basal forebrain (*Heckers et al., 1992*). In combination with these inputs, the diffuse projections of central thalamus allow it to influence the overall excitability of cortex during states of attention. For example, virtually all relay cells of the CL nucleus project to both striatum and cerebral cortex (*Deschenes et al., 1996*).

Studies on the physiological properties of central thalamus also show that it is tightly coupled to arousal regulation. First, variations in the level of activity within the intralaminar nuclei are linked to changes in behavioral alertness (*Kinomura et al., 1996*; *Shirvalkar et al., 2006*; *Mair and Hembrook, 2008*; *Schiff et al., 2013*; *Giber et al., 2015*), including transitions during the normal sleep–wake cycle and acute cognitive enhancements such as improved working-memory and sustained attention (*Baker et al., 2012*). Similarly, lesions of the central thalamus can produce enduring cognitive impairments, including reduced attentional processing and memory (*Guberman and Stuss, 1983*; *Mair et al., 1998*; *Van Der Werf et al., 1999*; *Newman and Burk, 2005*), hypersomnolence (*Bassetti et al., 1996*), or even coma (*Castaigne et al., 1981*; *Plum, 1991*). Indeed, neuronal loss across central thalamus has been associated with severely disabled and vegetative patients following severe traumatic brain injury (*Maxwell et al., 2006*). In addition, electrical stimulation of the central thalamus at low frequencies is associated with absence seizures and behavioral arrest in animal models (*Hunter and Jasper, 1949*) and human subjects (*Velasco, 1996*). Human imaging studies have also found that anesthesia-induced loss of consciousness is associated with disrupted thalamocortical functional connectivity in regions consistent with the intralaminar nuclei (*Akeju, 2014*).

According to the mesocircuit hypothesis of forebrain dysfunction, the central thalamus, which has a strong activating role in driving cortical and striatal neurons (*Schiff, 2010*), is under tonic inhibition by GABAergic pallidal neurons (*Grillner et al., 2005*). This GABAergic population is itself inhibited by the striatal neurons driven by central thalamus, creating a positive feedback loop. Thus, when the thalamostriatal and thalamocortical projections from central thalamus are partially lost due to brain injury, it causes disinhibition of the pallidum and increased inhibition of the remaining central thalamus neurons, which further reduces cortical activation. While this down-regulation is predicted to have broad modulatory impact on global dynamics, deep brain stimulation (DBS) of central thalamus

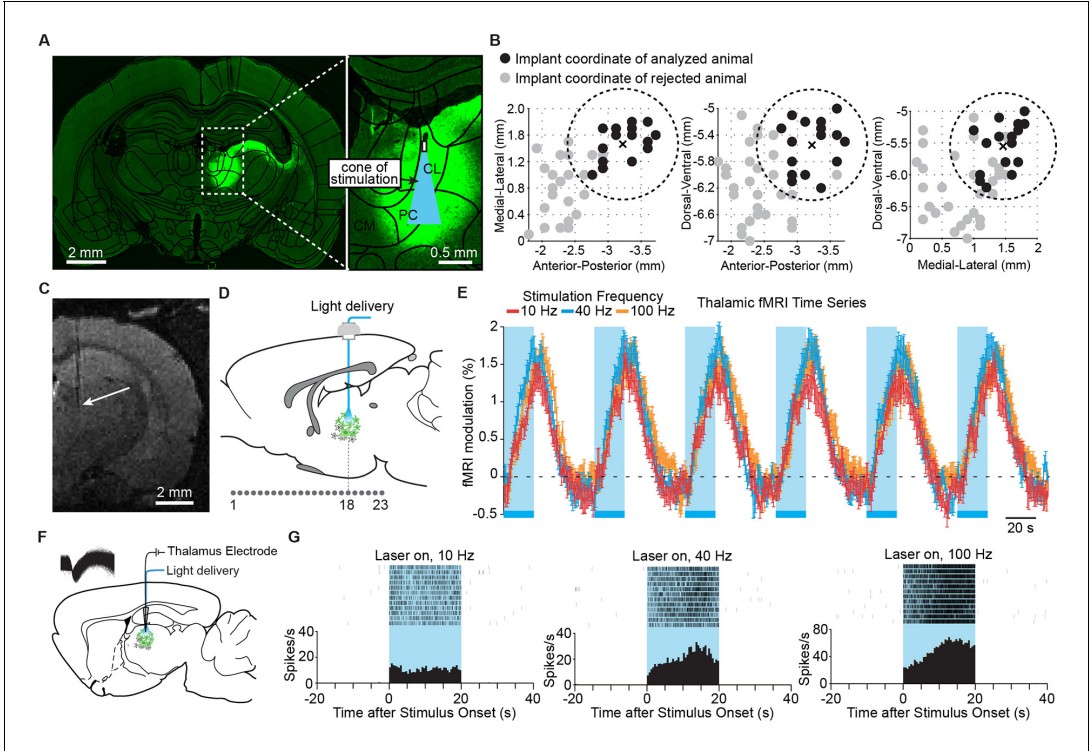

**Figure 1.** Targeted stimulation of central thalamus evokes positive BOLD changes and increases in neuronal firing at the site of stimulation. (**A**) Representative wide-field fluorescence image shows robust ChR2-EYFP expression throughout central thalamus, overlaid with the estimated cone of excited tissue shown to scale. (**B**) Empirically observed locations of fiber optic implants in initial cohort of 47 rats, estimated using high-resolution structural MRI scans. Of these animals, 18 had implant locations that were accurately localized to the central thalamus (<0.85 mm from target site, shown as dashed circle and cross). Two were excluded based on lack of thalamic activation, leaving n = 16 rats for further analysis. Black dots indicate implant coordinates of 16 animals used for analysis. Gray dots indicate implant coordinates of 31 rejected animals. (**C**) Representative T2-weighted anatomical MRI scan used to estimate implant location, marked with arrow. (**D**) Schematic of 23 coronal slices acquired during ofMRI experiments. Slice numbers correspond to activation maps in *Figure 2*. (**E**) Average time series of significantly modulated voxels within the ipsilateral thalamus ROI (see *Figure 2D*) exhibit robust positive BOLD responses during repeated 20 s periods of stimulation at 10, 40, and 100 Hz, indicated by blue bars. Values are mean ± s.e.m. across animals (n = 16, 10, and 16 for each frequency, respectively). (**F**) Diagram of local in vivo optrode recordings during optical stimulation of central thalamus. Inset shows spike waveforms of a recorded neuron. (**G**) Representative peri-event time histogram of a recorded neuron showing the increase in firing rate within central thalamus during optical stimulation at each of the three frequencies tested. See also *Figure 1—figure supplement 1* and *Figure 1—source data 1*. BOLD: Blood-oxygen-level-dependent; ROI: Regions of interest.

The following source data and figure supplement are available for figure 1:

**Source data 1.** Firing rates before, during, and after repeated 20 s stimulation periods for each of the five neurons recorded in central thalamus.

**Figure supplement 1.** Specificity of ChR2 targeting for CaMKIIa-positive cells.

has been explored as a potential means of reversing its effects and facilitating arousal regulation in the minimally conscious state (*Shirvalkar et al., 2006*; *Schiff et al., 2007*; *Mair and Hembrook, 2008*; *Smith et al., 2009*; *Mair et al., 2011*; *Schiff, 2012*; *Fridman et al., 2014*; *Gummadavelli et al., 2015*). Despite success in a single-subject clinical study (*Schiff et al., 2007*), identification of circuit-level mechanisms that link therapeutic efficacy of central thalamus DBS to specific stimulation parameters remains challenging and at present limits the clinical efficacy of DBS in subjects with traumatic brain injury. Thus, while significant progress has been made in understanding the connections of central thalamus and its behavioral correlates (*Van der Werf et al., 2002*; *Schiff, 2008*), relatively little is known about the dynamic function of these circuits, and no clear mechanism exists to explain how – or indeed, if – a single population in central thalamus can act as a switch on the global brain state.

To overcome such obstacles and dissect the dynamic influence of central thalamus on global brain networks, we combined targeted optogenetic control of excitatory relay neurons within central thalamus with whole-brain fMRI readouts, EEG, and single-unit recordings. The combination of optogenetic stimulation with fMRI (ofMRI) has been demonstrated to be an effective method for mapping the functional role of specific genetically- and spatially-defined neuronal populations at various brain regions and different frequencies of stimulation (*Lee et al., 2010*; *Desai et al., 2011*; *Weitz et al., 2015*; *Liang et al., 2015*; *Takata et al., 2015*; *Duffy et al., 2015*; *Byers et al., 2015*). In the present work, we sought to apply this approach to uncover the downstream effects of distinct firing patterns by central thalamus relay cells at the whole-brain level – a visualization uniquely possible with ofMRI. Prior work beginning with Morison and Dempsey (*Morison and Dempsey, 1942*), who envisioned "dissecting… the electrical activity of the cortex on the basis of its relations with the thalamus," has shown that high- and low-frequency electrical stimulation of thalamic nuclei can produce distinct cortical activation patterns. For example, early studies of the intralaminar nuclei per se showed that low-frequency stimulation evokes slow-wave activity and spindle bursts in cortical EEG, which are associated with primary generalized absence seizures, loss of consciousness, and drowsiness (*Hunter and Jasper, 1949*; *Jasper, 1949*; *Seidenbecher and Pape, 2001*). Conversely, high-frequency electrical stimulation has been shown to desynchronize the cortical EEG signal (*Moruzzi and Magoun, 1949*), which is associated with behavioral arousal. While these studies set early hypotheses on the mechanisms of arousal regulation, the non-selective nature of electrical stimulation has prevented the observed responses from being attributed specifically to relay cells, and not synaptic afferents or fibers of passage mixing together in a bulk activation effect. More importantly, although a picture of whole brain activity can be vaguely inferred from electrophysiology recordings, these techniques cannot provide a direct visualization of activity across individual brain regions over the entire brain. Because ofMRI can provide spatial and temporal information on the whole-brain scale during perturbations of specific neural circuitry, we applied this technique to study the causal role of central thalamus relay neurons in activating forebrain networks.

In addition, following on novel results described below, we were led to examine the interplay between central thalamus and zona incerta (ZI), a subcortical region implicated in the modulation of absence seizures in rats (*Shaw et al., 2013*). The ZI, a mostly GABAergic region, has been shown to limit the transmission of ascending sensory information via feedforward inhibition of higher order thalamic nuclei (*Barthó et al., 2002*; *Trageser and Keller, 2004*; *Lavallée et al., 2005*; *Trageser et al., 2006*). Such activity can induce a state of reduced sensory processing, similar to the behavioral quiescence induced by low-frequency central thalamus stimulation. These studies suggest a powerful control exerted by ZI over brain state and higher level processing, much like central thalamus. However, the possible involvement of ZI in central thalamus arousal circuits remains unexplored. Here, we investigated the electrophysiology responses of this region during targeted stimulation of central thalamus at frequencies that either facilitate or suppress attention and arousal. To determine whether ZI plays an active role in these circuits, we also used optogenetics to specifically inhibit neurons in this region during central thalamus stimulation. This experimental paradigm was used to infer ZI's functional contribution to the central thalamus-driven brain circuit dynamics measured with fMRI and electrophysiology.

## Results

### High-frequency stimulation of central thalamus relay neurons drives widespread forebrain activation in vivo

To investigate the specific role of central thalamus, we applied optogenetic techniques to control relay cells in a spatially and temporally precise manner. We performed a stereotactic injection in the right CL and PC intralaminar nuclei of central thalamus with adeno-associated virus carrying channelrhodopsin-2 (ChR2) and the fluorescent reporter protein EYFP under control of the CaMKIIa promoter. This promoter is expressed primarily in excitatory neurons, the vast majority of which in thalamus are relay cells (*Smith, 2008*; *Ellender et al., 2013*). Of cells identified within the bulk injection area, 35% were EYFP-positive, and 97% of EYFP-positive cells co-expressed CaMKIIa, indicating high sensitivity for stimulation of excitatory neurons (n = 2 rats, 831 cells; *Figure 1—figure supplement 1*). While ChR2-EYFP expression extended beyond these two nuclei (*Figure 1A*), targeted

stimulation of the intralaminar nuclei was achieved by (a) stereotactic placement of the implanted optical fiber, as confirmed with high-resolution T2-weighted structural MR images (*Figure 1B,C*), and (b) spatially restricted illumination (*Figure 1A,B*). We initially injected and cannulated 47 rats using the central thalamus as the stereotactic target (-3.2 mm AP, +1.5 mm ML, -5.5 mm DV). However, the intralaminar nuclei are relatively small and difficult to accurately target. We therefore used only a subset of these animals based on the empirically observed distribution of optical fiber tip locations using T2-weighted MRI scans (*Figure 1B*; <0.85 mm distance from target coordinate). Of the 18 rats that had an accurately localized implant location, two exhibited a general absence of fMRI activity – most notably at the site of stimulation – and were excluded, leaving 16 animals for further analysis.

In order to achieve a small volume of directly excited tissue limited to the intralaminar nuclei, we used a 62.5-µm diameter optical fiber. Assuming that an intensity of 1 mW/mm$^2$ is required for ChR2 activation (*Aravanis et al., 2007*), the specific power exiting from the fiber optic's tip in these experiments (2.5 mW) corresponds to a penetration depth of 1.08 mm and a volume of 0.08 mm$^3$ over which ChR2+ neurons can be excited. *Figure 1A* illustrates this penetration depth and activation cone (11.7° half-angle of divergence) to scale with the targeted nuclei, showing that stimulation is well restricted to the central thalamus. These two factors (MR-validated stereotactic fiber placement and a small volume of excited tissue) suggest that the effects reported here primarily derive from stimulation of excitatory relay neurons within the central thalamus.

To explore the anatomical connectivity of transfected neurons in central thalamus, we collected ex vivo fluorescence microscopy images of ChR2-EYFP expression. Due to the spread of viral transfection (*Figure 1A*), it is possible that the reported fluorescence reflects projections from adjacent thalamic nuclei as well. Nevertheless, in agreement with known projection systems of central thalamus, EYFP-expressing axons were observed throughout forebrain, including frontal cortex and striatum (*Figure 2—figure supplement 1*). In particular, the medial prefrontal, lateral prefrontal, cingulate, motor, and sensory cortices all received strong projections. This input was highly convergent at the superficial layers, with moderate but weaker projections present in middle layers as well. Furthermore, projections were significantly restricted to the hemisphere ipsilateral to virus injection for both cortex and striatum. While these anatomical connections provide a strong foundation for understanding how central thalamus can influence brain state, they do little to explain the dynamic nature of these circuits – for example, how stimulation of central thalamus at different frequencies can lead to distinct behavioral responses (*Hunter and Jasper, 1949*; *Velasco, 1996*). Therefore, to dissect the functional significance of these massive forebrain projections and visualize the large-scale spatial and temporal dynamics evoked by central thalamus stimulation, we combined optical stimulation with simultaneous in vivo whole-brain functional imaging (*Lee et al., 2010*).

During optogenetic fMRI experiments, 23 coronal slices with 0.5 × 0.5 mm$^2$ in-plane resolution and 0.5 mm thickness were acquired at a frame rate of 750 ms using spiral k-space trajectories and a sliding window reconstruction algorithm to achieve high-spatiotemporal resolutions with whole-brain coverage (bregma +5.2 to -5.3 mm; *Figure 1D*) (*Fang and Lee, 2013*). Novel inverse Gauss-Newton methods were also used to correct for possible motion artifacts and optimize the robustness of detecting optogenetically evoked responses (*Fang and Lee, 2013*). For each experiment, we delivered 20 s periods of stimulation every minute for 6 min at 10, 40, or 100 Hz. This form of continuous steady-state stimulation mimics the approach used in clinical DBS and has been showed to evoke robust fMRI responses with optogenetic stimuli (*Lee, et al., 2010*; *Duffy, et al., 2015*; *Weitz et al., 2015*). Indeed, stimulation at all three frequencies resulted in a robust positive blood-oxygen-level-dependent (BOLD) signal at the site of stimulation that was highly synchronized to light delivery, increased upon optical activation, and gradually returned to baseline following the end of stimulation (*Figure 1E*). To confirm that this BOLD signal reflected underlying neuronal firing patterns, we next performed simultaneous single-unit recordings with stimulation using an optrode at the central thalamus (*Figure 1F*). In agreement with the fMRI signal, stimulations at 10, 40, and 100 Hz all resulted in robust increases in the local neuronal firing rate (*Figure 1G;* n = 5 neurons, p< 0.05, Wilcoxon signed-rank test between the 20 s pre-stimulation and stimulation periods, 12 trials for each neuron).

Both locally in the thalamus and at downstream, synaptically connected brain regions, the frequency of stimulation was a critical parameter in determining the extent of ipsilateral and contralateral BOLD activation – defined here as positive BOLD signals significantly synchronized to the block

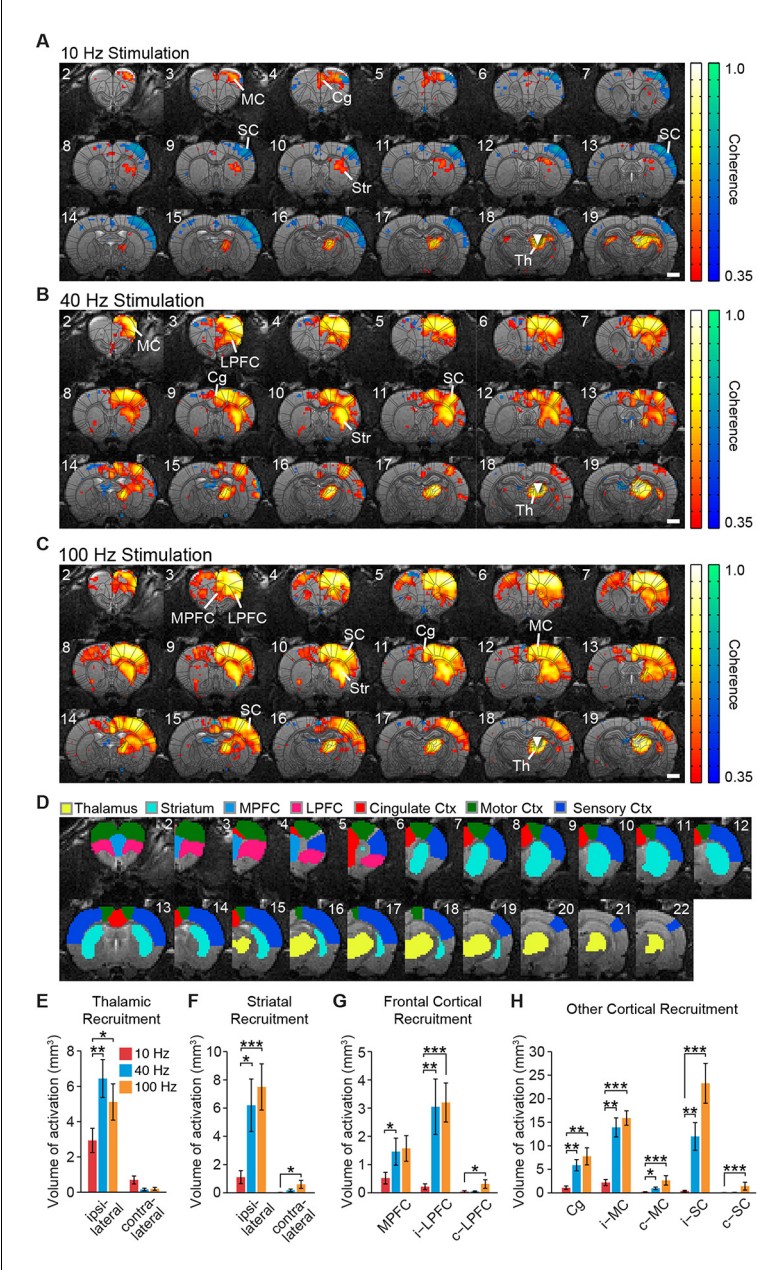

**Figure 2.** Spatial characterization of evoked fMRI signals. (A–C) Average coherence maps of brain-wide activity during stimulation of excitatory central thalamus relay neurons at 10, 40, and 100 Hz. Warm colors indicate positive BOLD responses, while cool colors indicate negative BOLD responses (see 'Materials and methods'). (D) Regions of interest (ROIs) used for quantitative analysis of spatial ofMRI activation patterns. (E) The amount of active volume (positive signal with coherence > 0.35) in the ipsilateral thalamus is significantly greater during 40 and 100 Hz stimulations than 10 Hz stimulation. Thalamic recruitment is relatively limited on the contralateral side. (F) Activation of the ipsilateral striatum is significantly greater during 40 and 100 Hz stimulations than 10 Hz stimulation. Activation of the contralateral striatum is limited across frequencies, although there is an increase from 10 to 100 Hz. (G) Medial and lateral prefrontal cortex exhibit a significantly greater volume of activation during 40 and/or 100 Hz stimulation than 10 Hz stimulation. Activity in the contralateral hemisphere is limited across all tested frequencies, although there is an increase from 10 to 100 Hz. (H) Activation of cingulate, motor, and somatosensory cortex is each greater during 40 and 100 Hz stimulations than 10 Hz stimulation. The contralateral motor and sensory cortices are also activated to a greater extent during 40 and/or 100 Hz stimulation. Scale bars in panels **A** through **C** represent 2 mm. Asterisks in panels **E** through **H** indicate significant differences for 10 versus 40 Hz and 10 versus 100 Hz stimulations. *p < 0.05, **p < 0.005, ***p < 0.001, one-sided Wilcoxon signed-rank tests, corrected for multiple comparisons. Error bars represent mean ± s.e.m. across animals. n = 16, 10, and 16 animals for 10, 40, and 100 Hz, respectively. Abbreviations are as follows: i- (ipsilateral), c- (contralateral), Cg (cingulate cortex), MC (motor cortex), MPFC (medial prefrontal cortex), LPFC (lateral prefrontal cortex), SC (sensory cortex), Str (striatum), Th (thalamus). See also *Figure 2—figure supplements 1/2/3* and *Figure 2—source data 1*.

*Figure 2 continued on next page*

*Figure 2 continued*

The following source data and figure supplements are available for figure 2:

**Source data 1.** Animal-specific activation volumes for the twelve regions of interest shown in *Figure 2E–H*.

**Figure supplement 1.** Representative fluorescence images of ChR2-EYFP at remote targets illustrate the massive projections to forebrain from transfected relay neurons in the right central thalamus.

**Figure supplement 2.** Widespread and frequency-dependent recruitment of forebrain with optogenetics is distinct to stimulation of intralaminar nuclei of central thalamus.

**Figure supplement 3.** The frequency-dependent recruitment of forebrain by central thalamus and its control over cortical BOLD signal polarity are preserved when pulse width is held constant.

stimulation paradigm (see 'Materials and methods'). In general, a much larger volume of brain tissue was activated by stimulation at 40 and 100 Hz compared to 10 Hz, with frontocortical areas and striatum being strongly activated at high frequencies (*Figure 2A–C*; *Videos 1–3*). To quantify these spatial differences in recruitment patterns, we calculated the total volume of positive and statistically significant BOLD signals evoked by stimulation in select region of interests (ROIs) (*Figure 2D*). This difference in activation volume between low- (10 Hz) and high- (40 or 100 Hz) stimulation frequencies was significant at the thalamus, striatum, and medial prefrontal, lateral prefrontal, cingulate, motor, and sensory cortices (*Figure 2E–H*). Striatal activity was primarily localized to the dorsal sector, with negligible activity occurring in the ventral region (*Figure 2B,C*). Furthermore, BOLD activation was generally restricted to the ipsilateral hemisphere, although activation volumes in the contralateral striatum, lateral prefrontal cortex, motor cortex, and sensory cortex were all significantly greater during 100 Hz stimulation compared to 10 Hz stimulation (*Figure 2F–H*).

These results provide a direct, region-specific visualization of the widespread driving effect that central thalamus has been shown to exert over forebrain, and link prior anatomical and physiological studies on arousal regulation to spatially precise and quantitative measures of cortical and striatal activation. For example, the evoked responses are consistent with the unilateral nature of thalamocortical projections (*Figure 2—figure supplement 1*), but reveal that the contralateral cortex can still be modulated by unilateral stimulation of central thalamus, particularly at high frequencies. Importantly, stimulation of other thalamic nuclei failed to evoke similarly widespread activity in the striatum and cortex (*Figure 2—figure supplement 2A*). Furthermore, large differences in forebrain activation between 10 and 40 Hz stimulations were not observed for other forms of subcortical stimulation (*Figure 2—figure supplement 2B*), suggesting this is a distinct property of central thalamus.

Throughout these experiments, a constant duty cycle of 30% was used to maintain the total amount of light delivery across frequencies and control for possible heating artifacts (*Christie et al., 2013*). Because we wished to keep a 20 s pulse train for all stimulation frequencies and avoid possible differences introduced by neuronal adaptation, maintaining a constant duty cycle required unique pulse widths for each frequency (i.e. 30, 7.5, and 3 ms for 10, 40, and 100 Hz, respectively). To rule out the possibility that these changes in pulse width were the primary cause of the above differences in forebrain recruitment, we repeated stimulations while maintaining a 3 ms pulse width. Visualization and quantification of evoked fMRI responses show that the increase in cortical and striatal activation with frequency was preserved (*Figure 2—figure supplement 3A,B*). These data suggest that stimulation frequency was the primary factor in modulating forebrain fMRI activation.

## Central thalamus stimulation frequency controls cortical excitation/inhibition balance

We next examined the temporal dynamics of cortical responses evoked during low- and high-frequency central thalamus stimulation. Despite targeted activation of excitatory neurons, the somatosensory cortex exhibited a strong negative BOLD signal during 10 Hz stimulation, suggesting a suppression of baseline activity (*Figures 2A* and *3A,B*). In stark contrast, central thalamus stimulations at 40 and 100 Hz led to positive changes in the BOLD signal at the somatosensory cortex

**Video 1.** Spatiotemporal dynamics of ofMRI activity during 10 Hz stimulation of excitatory relay neurons of the central thalamus. Highlighted voxels are restricted to those significantly synchronized to the block stimulation paradigm, as determined by frequency domain analysis. Color coding reflects the instantaneous relative percent modulation of each voxel's hemodynamic response function, thresholded over ± (0.2 to 1.5)%. Laser status indicates the 20 s period of stimulation (2–22 s). Abbreviations are as follows: SC (sensory cortex), Th (thalamus).

(*Figures 2B,C* and *3A,B*). Thus, stimulation of the same excitatory population at different frequencies resulted in completely opposite responses at a downstream target. Importantly, these responses were preserved when pulse width was held constant in control experiments, indicating that stimulation frequency was the primary factor controlling this effect (*Figure 2—figure supplement 3A,C*).

While previous studies have hinted at similar findings of frequency-dependent polarity changes (*Logothetis et al., 2010*; *Weitz et al., 2015*), downstream positive and negative BOLD signals that result from selective stimulation of excitatory neurons at distinct frequencies have not yet been visualized and validated with electrophysiology. To define the neuronal underpinnings of these signals, we therefore performed single-unit extracellular recordings in the somatosensory cortex during central thalamus stimulation (*Figure 3C*). In agreement with the BOLD activity observed during ofMRI experiments, 10 Hz stimulation resulted in a decrease in neuronal firing rate between pre-stimulation and stimulation periods (*Figure 3D,E*; n = 10 of 11 recorded neurons). Conversely, stimulations at 40 and 100 Hz both led to increases in neuronal firing (*Figure 3D,E*; n = 11 of 11 recorded neurons). Because the evoked firing rates appeared to change over the course of stimulation, we specifically compared the pre-stimulation firing rate to the average firing rates during consecutive 5 s periods of the 20 s stimulus (i.e. 0–5 s, 5–10 s, 10–15 s, and 15–20 s; uncorrected p < 0.05, Wilcoxon signed rank test; 17 trials for each neuron). Interestingly, the decrease in firing rate during 10 Hz stimulation occurred primarily over the interval from 5 to 15 s after stimulation began, while the increase in firing rate during 40 Hz stimulation occurred primarily over the first 10 s (*Table 1*). On the other hand, the increase in neuronal firing rate during 100 Hz stimulation was generally maintained throughout the 20 s stimulation period (*Table 1*). Such differences may reflect short-term plasticity of the thalamocortical pathway, which has previously been shown to exhibit frequency-dependent properties (*Castro-Alamancos and Connors, 1996a*; *1996b*). Peri-stimulus time histograms also revealed that spike events occurring during inhibition had a non-uniform distribution over time, which peaked between 6 and 34 ms after light onset (*Figure 3—figure supplement 1*). These data suggest that the glutamatergic thalamocortical input at 10 Hz sometimes generated action potentials. Notably, however, not every light pulse resulted in an immediate action potential.

Together, these ofMRI and electrophysiological data indicate that neuronal activity throughout somatosensory cortex is suppressed at low frequencies of central thalamus stimulation and increased at high frequencies of stimulation. Because our stimulations were restricted to excitatory neurons with cell bodies located in central thalamus, the causal relationship between stimulation frequency and cortical excitation/inhibition can be attributed to the neurons' initial firing pattern. These results add to a growing body of literature in systems neuroscience suggesting that a neuronal population's firing pattern can have vastly different – even opposite – effects on downstream regions depending on its specific temporal code (*Dempsey and Morison, 1943*; *Logothetis et al., 2010*; *Mattis et al., 2014*; *Weitz et al., 2015*).

**Video 2.** Spatiotemporal dynamics of ofMRI activity during 40 Hz stimulation of excitatory relay neurons of the central thalamus. Highlighted voxels are restricted to those significantly synchronized to the block stimulation paradigm, as determined by frequency domain analysis. Color coding reflects the instantaneous relative percent modulation of each voxel's hemodynamic response function, thresholded over ± (0.2 to 1.5)%. Laser status indicates the 20 s period of stimulation (2–22 s). Abbreviations are as follows: Cg (cingulate cortex), LPFC (lateral prefrontal cortex), MC (motor cortex), MPFC (medial prefrontal cortex), SC (sensory cortex), Str (striatum), Th (thalamus).

**Video 3.** Spatiotemporal dynamics of ofMRI activity during 100 Hz stimulation of excitatory relay neurons of the central thalamus. Highlighted voxels are restricted to those significantly synchronized to the block stimulation paradigm, as determined by frequency domain analysis. Color coding reflects the instantaneous relative percent modulation of each voxel's hemodynamic response function, thresholded over ± (0.2 to 1.5)%. Laser status indicates the 20 s period of stimulation (2–22 s). Abbreviations are as follows: Cg (cingulate cortex), LPFC (lateral prefrontal cortex), MC (motor cortex), MPFC (medial prefrontal cortex), SC (sensory cortex), Str (striatum), Th (thalamus).

## Low-frequency central thalamus stimulation drives incertal oscillations

Given that stimulation was restricted to excitatory neurons, we hypothesized that the suppression of cortex during 10 Hz stimulation might be related to the frequency-dependent modulation of a GABAergic population. We chose to investigate the response properties of the ZI, which has been implicated in providing a powerful GABAergic modulation of 10 Hz spike-wave activity in spontaneous absence seizures in the rat (*Shaw et al., 2013*). Anatomically, ZI sends direct GABAergic projections to somatosensory thalamic nuclei and sensory cortex (*Nicolelis et al., 1995*; *Kolmac and Mitrofanis, 1999*; *Barthó et al., 2002*). Functionally, ZI has also been shown to selectively gate sensory information processing in higher order thalamic nuclei through GABAergic inhibition (*Trageser and Keller, 2004*; *Lavallée et al., 2005*; *Trageser et al., 2006*). To investigate the involvement of ZI, we performed single-unit and field potential electrophysiology recordings in this region during simultaneous optogenetic stimulation of central thalamus at 10 or 40 Hz (*Figure 4A*). EEG recordings were simultaneously collected in frontal cortex to directly evaluate the relationship between ZI activity and whole-brain arousal state, which is typically measured with forebrain EEG. The ZI was targeted using stereotactic localization and the well-defined somatotopic representation of this region (*Nicolelis et al., 1992*). The electrode was targeted to -3.96 mm AP, +2.2–2.6 mm ML, +6.7–7.2 mm DV from dura. The ZI was identified according to a compatible depth reading, spike latencies consistent with a polysynaptic response (on the order of 10 ms; *Figure 4B*), and a receptive field that responds to contralateral whisker stimulation, which ZI is known to possess (*Nicolelis et al., 1992*). The electrode was initially lowered through the dorsal part of the VP thalamus (approximately 1.5 mm above ZI), which also responds to whisker stimulation, until the recorded neurons did not respond to such a stimulus. The electrode was then lowered for another ~1.5 mm until the recorded neurons fired in response to whisker stimulation, indicating the ZI had been reached.

Out of 28 isolated ZI neurons, the majority exhibited increases in their firing rate during central thalamus stimulation at both 10 and 40 Hz (*Figure 4C*; n = 26 and 22, respectively; p < 0.05, Wilcoxon signed rank test between the 20 s pre-stimulation and stimulation periods, 20 trials for each neuron). However, a key difference was that large, amplitude-modulated spindle-like oscillations (SLOs) in the field potential occurred significantly more often during 10 Hz stimulation than 40 Hz stimulation (*Figure 4D,E*). These oscillations exhibited an inter-event interval centered around 6.6 ± 0.2 s (s.e.m.), similar to those observed in thalamus during sleep onset (*Contreras et al., 1997*) (*Figure 4F*). Consistent with this, simultaneous EEG recordings in frontal cortex revealed strong spike-wave modulation during 10 Hz stimulation and lower amplitude, fast oscillations during 40 Hz stimulation, which are associated with loss of consciousness and aroused brain states, respectively (*Figure 4G*). EYFP-expressing axons were also observed in ZI (*Figure 4H*), indicating that central thalamus relay neurons may have direct connections to ZI and providing a possible anatomical substrate for these responses.

## Cortical inhibition driven by central thalamus stimulation depends on evoked incertal activity

The observation of spindle-like oscillations in ZI during 10, but not 40, Hz central thalamus stimulation indicates that this region can be uniquely engaged by central thalamus-driven networks. However, it remains unknown whether the evoked activity in ZI plays a causal role in driving the frequency-dependent inhibition of somatosensory cortex. To address this question, we injected the

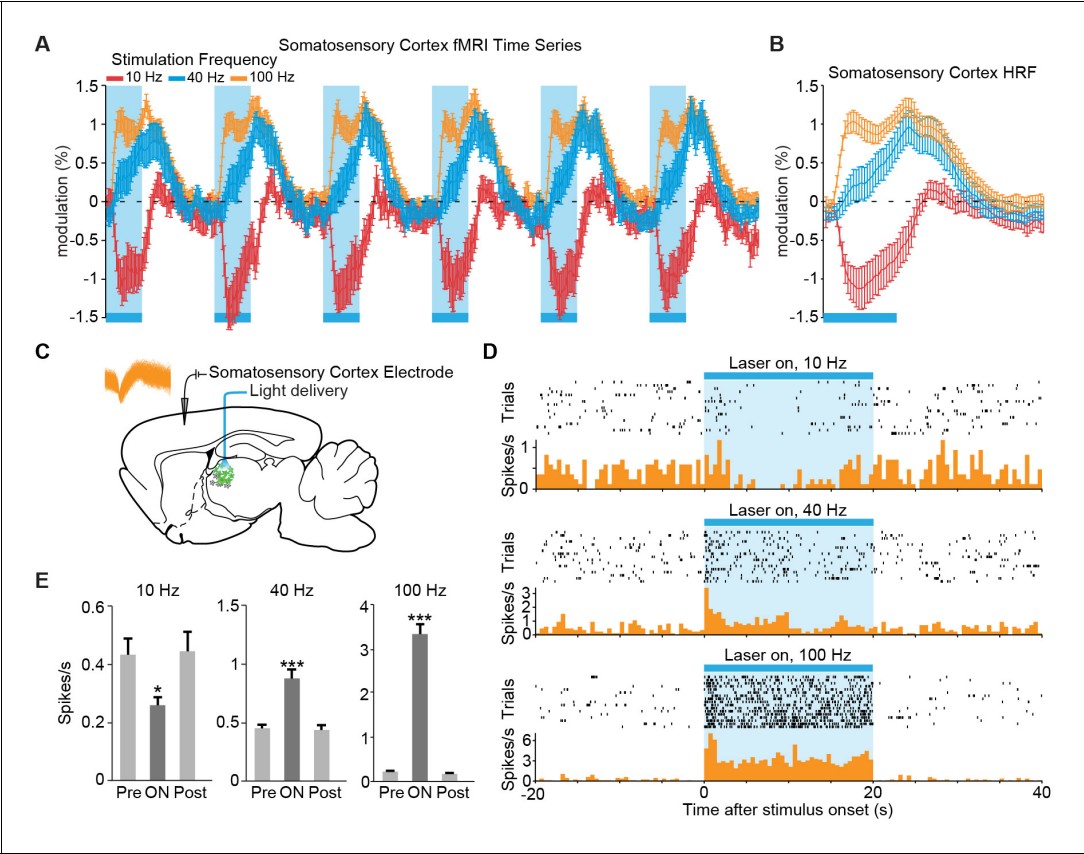

**Figure 3.** The sign of evoked cortical activity depends on the frequency of central thalamic stimulation. (A,B) 10 Hz stimulation of central thalamus evokes a strong negative BOLD signal throughout ipsilateral somatosensory cortex, while 40 and 100 Hz stimulations evoke positive responses. Time series come from the sensory cortex ROI defined in *Figure 2D*. Hemodynamic response function (HRF) shows the average response to a single 20 s period of stimulation, indicated by blue bar. Error bars represent mean ± s.e.m. across animals. n = 16, 10, and 16 for 10, 40, and 100 Hz, respectively. (C) Diagram of in vivo recordings at somatosensory cortex during stimulation of excitatory central thalamus relay neurons. Inset shows spike waveforms of a recorded neuron. (D,E) Representative peri-event time histogram of a recorded neuron, and corresponding quantification of firing rate during the 20 s periods before, during, and after stimulation. Neural firing rate decreased within the somatosensory cortex during 10 Hz central thalamus stimulation, but increased during 40 and 100 Hz stimulations (n = 17 trials each, *p < 0.05, ***p < 0.001 pre vs. ON, two-tailed Wilcoxon signed-rank test; see *Table 1* for further analysis). Values are mean ± s.e.m. See also *Figure 2—figure supplement 3C* and *Figure 3—figure supplement 1*.

The following figure supplement is available for figure 3:

**Figure supplement 1.** Cortical spikes that occur during periods of inhibition driven by 10 Hz central thalamus stimulation exhibit a non-uniform distribution over time.

inhibitory opsin halorhodopsin (eNpHR) fused to the mCherry fluorescent marker and controlled by the pan-neuronal hSyn promoter into ZI of four animals expressing ChR2-EYFP in central thalamus (*Figure 5A,B*, *Figure 5—figure supplement 1*). Two new stimulation paradigms were explored: (1) 20 or 30 s continuous eNpHR activation, and (2) 20 s, 10 Hz central thalamus stimulation performed within a 30 s period of continuous eNpHR activation. Single-unit recordings were performed simultaneously at the ZI and sensory cortex during concurrent activation of these two opsins (*Figure 5C*).

Among the 70 neurons recorded in ZI, delivery of 593 nm light resulted in a decrease in firing for 62 cells (p < 0.05, Wilcoxon signed rank test between 20 s pre-stimulation period and 20 or 30 s stimulation period, 15–20 trials for each neuron), indicating that illumination of halorhodopsin was successful in suppressing incertal activity. The evoked decrease in neuronal firing rate typically lasted throughout the duration of 593 nm light delivery (*Figure 5D*). When halorhodopsin activation in ZI was paired with 10 Hz stimulation of central thalamus, the previously described increase in incertal firing (*Figure 4C*) was disrupted. In 60 out of 70 neurons, the difference in incertal firing rate

**Table 1.** Electrophysiology results from sensory cortex single-unit recordings. See also *Table 1—source data 1*.

| Stimulation frequency | Effect on sensory cortex firing rate | Percentage of neurons with significant change in firing rate (n = 11) | | | |
|---|---|---|---|---|---|
| | | 0–5 s after stim. onset | 5–10 s after stim. onset | 10–15 s after stim. onset | 15–20 s after stim. onset |
| 10 Hz | Increase | 0% | 0% | 0% | 0% |
| | Decrease | 0% | 91% | 82% | 9% |
| 40 Hz | Increase | 100% | 91% | 36% | 55% |
| | Decrease | 0% | 0% | 0% | 0% |
| 100 Hz | Increase | 100% | 82% | 82% | 82% |
| | Decrease | 0% | 0% | 0% | 0% |

Source data 1. Firing rates before, during, and after repeated 20 s stimulation periods for each of the 11 neurons recorded in somatosensory cortex. Exact p values comparing the somatosensory cortex firing rate before and during stimulation are provided. The 20 s stimulation period was divided into four consecutive 5 s blocks to evaluate the change in firing rate over time.

between the 20 s 10 Hz central thalamus stimulation period and the pre-stimulation period was significantly reduced with concurrent eNpHR activation (*Figure 5F*; $p < 0.05$, one-sided Wilcoxon rank sum test, n = 10–20 trials). *Figure 5E* illustrates the suppression of ZI activity throughout the 20 s period of 10 Hz central thalamus stimulation in a representative neuron. These data indicate that activation of halorhodopsin significantly suppressed the incertal firing evoked by 10 Hz central thalamus stimulation, and successfully disrupted incertal processing.

To determine whether this suppression of ZI affected the cortical activity driven by central thalamus stimulation, we quantified the changes in somatosensory cortex firing rate evoked by ChR2 activation with and without illumination of eNpHR. Seventy-six somatosensory cortex neurons were recorded, and the 20 s period of central thalamus stimulation was divided into four 5 s bins as before. Consistent with the data presented in *Figure 3*, 68 cells (89%) exhibited a decrease in firing during 10 Hz stimulation of central thalamus (uncorrected $p < 0.05$, Wilcoxon signed rank test; 10–15 trials for each neuron). Strikingly, however, suppression of ZI activity with eNpHR reversed this effect. Across animals, 50 out of 76 neurons (66%) exhibited reduced inhibition when central thalamus stimulation was paired with eNpHR activation (*Figure 5H*; $p < 0.05$, Wilcoxon rank sum test over 1 s bins; 10–20 trials for each neuron). Indeed, a fraction of cells switched from inhibitory responses to excitatory ones. *Figure 5G* illustrates the firing patterns of one cell that exhibited an inhibitory response during 10 Hz central thalamus stimulation that was eliminated when ZI was simultaneously suppressed with eNpHR. Collectively, these data suggest that incertal activity during 10 Hz central thalamus stimulation has a net inhibitory effect on somatosensory cortex. In support of this influence being through direct anatomical connections, mCherry-positive axons were observed in the sensory cortex (*Figure 5I*), consistent with previous reports of incerto-cortical projections (*Lin et al., 1990*). These findings present a conceptually novel role of ZI in central thalamus arousal circuits.

## Central thalamus stimulation modulates brain state in a frequency-dependent manner

Finally, to relate these findings more directly to behavior associated with central thalamus arousal circuits and previous electrical stimulation studies, we performed 10, 40, and 100 Hz stimulations in asleep, unanaesthetized animals with simultaneous video and EEG recordings (see 'Materials and methods'). Control (pre-stimulus) activity was consistent across frequencies of stimulation, as quantified with EEG band power in delta, theta, alpha, and beta bands (*Figure 6—figure supplement 1*). During 10 Hz stimulation, the majority of animals exhibited behavior indicative of an absence seizure, including freezing and behavioral arrest throughout stimulation followed by a return to sleep (*Figure 6A*; n = 4/7). In addition, the most common EEG response was a transition to slow spike-wave discharges (*Figure 6B,C*; n = 5/7), which are typically associated with loss of consciousness (*Mirsky and VanBuren, 1965*). In stark contrast, stimulations at 40 and 100 Hz resulted in

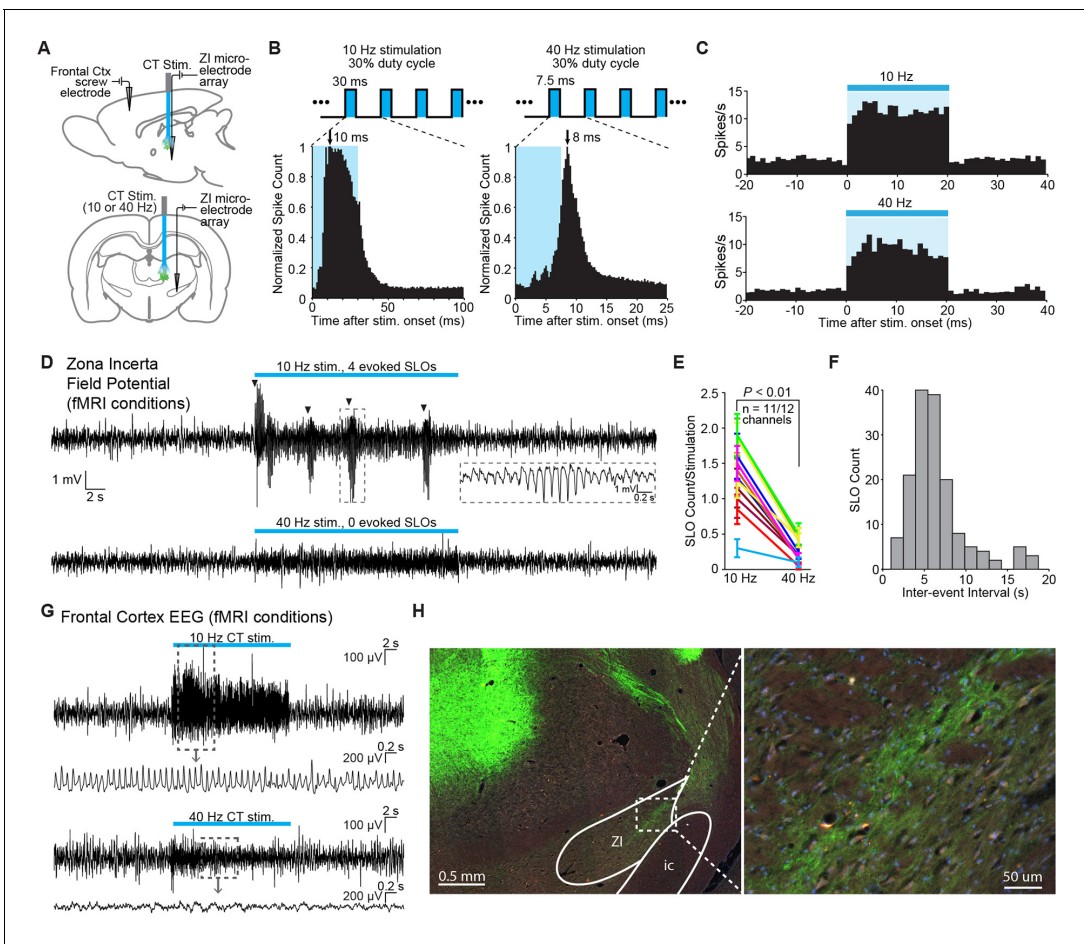

**Figure 4.** Frequency-dependent spindle-like oscillations are evoked in zona incerta (ZI). (**A**) Diagram of in vivo recordings at ZI and simultaneous EEG recordings in frontal cortex during optical stimulation of central thalamus in anesthetized animals. (**B**) Representative peri-event time histograms of spiking activity from recorded ZI neurons aligned to the onset of individual light pulses, summed over all pulses and trials. Peak spike latencies are approximately 10 and 8 ms for 10 Hz (left) and 40 Hz (right) stimulations, suggesting that recordings are performed at least one synapse away from the stimulated population in central thalamus. Schematics at top illustrate the 30% duty cycle pulse trains which lasted 20 s for each frequency. (**C**) Representative peri-event time histograms over the 20 s period of stimulation show increases in ZI firing during 10 and 40 Hz stimulations. Among the 28 isolated single-units in ZI (n = 2 animals), most exhibited a significant increase in firing rate during stimulation (n = 26 and 22 out of 28 neurons, respectively; p < 0.05, one-tailed Wilcoxon signed-rank test with 20 trials for each cell). (**D**) Representative field potential recordings from the same channel and trial number during 10 Hz (top) and 40 Hz (bottom) stimulation of central thalamus. Four amplitude-modulated, spindle-like oscillations (SLOs) are evoked during 10 Hz stimulation (marked by black triangles), while none are evoked during 40 Hz stimulation. Inset shows a zoomed-in SLO. (**E**) The number of SLOs was greater during 10 Hz stimulation than 40 Hz stimulation across 11 of 12 available channels (n = 2 animals, 20 trials each, p < 0.01, one-tailed Wilcoxon rank sum test). (**F**) When more than one SLO was evoked within the same 20 s period of 10 Hz stimulation, the distribution of inter-event intervals was centered at 6.6 ± 0.2 s (s.e.m.). (**G**) Representative EEG recordings collected in frontal cortex during central thalamus stimulation and simultaneous ZI recordings. 10 Hz stimulation evokes a spike-wave response, which is associated with loss of consciousness and perceptual awareness, while 40 Hz stimulation evokes a low voltage fast response indicative of arousal. (**H**) ChR2-positive processes were observed in ZI, providing a basis for its recruitment during stimulation of central thalamus. i.c.: internal capsule. See also *Figure 4—source data 1*, *2*. EEG: Electroencephalography.

The following source data is available for figure 4:

**Source data 1.** Firing rates before, during, and after repeated 20 s central thalamus stimulation periods for each of the 28 neurons recorded in zona incerta.

**Source data 2.** Quantification of spindle-like oscillation (SLO) occurrences in the zona incerta during 10 and 40 Hz stimulation of central thalamus.

behavioral transitions to an awake state, reflected by exploration and goal-directed movement (*Figure 6A*; n = 4/7 and 4/6, respectively). Similarly, the most common EEG pattern evoked by these

high-frequency stimulations was a low voltage fast response (*Figure 6B*; n = 3/7 and 6/6, respectively), indicative of cortical activation and desynchronization. Collectively, these phenomena are consistent with the patterns of cortical and striatal recruitment observed with ofMRI. Moreover, the slow spike-wave and low voltage fast EEG responses evoked during behavioral experiments (*Figure 6C,D*) match those recorded under anesthetized conditions (*Figure 4G*), further linking the network activation patterns revealed by ofMRI to the arousal responses reported here, as well as those reported in early stimulation studies (e.g. [*Hunter and Jasper, 1949*]).

## Discussion

Previously proposed mechanisms of arousal regulation have focused on the physiological and anatomical specialization of neurons within central thalamus (*Schiff, 2008*). Here, using a combination of ofMRI and electrophysiological recordings, we directly visualize whole-brain network activations produced by selective stimulation of central thalamus relay neurons and reveal novel insight into frequency-dependent gating of forebrain arousal.

Since the earliest observations that electrical stimulation of central thalamus exerts frequency-dependent effects on behavior and EEG rhythms (*Moruzzi and Magoun, 1949*; *Hunter and Jasper, 1949*), behavioral arousal and cognition have been tightly linked with cortical activation (i.e. low-amplitude, high-frequency oscillations), while behavioral arrest has been linked with cortical deactivation (i.e. high-amplitude, low-frequency oscillations). More recently, it has also been shown that pharmacologically-induced changes in thalamic firing levels can switch cortical dynamics between activation and deactivation (*Hirata and Castro-Alamancos, 2010*). However, no studies have characterized the specific changes in activity that simultaneously occur across the whole intact brain during these events to explain how the same neuronal population can selectively switch arousal state. Here, we show that distinct firing patterns of excitatory neurons in the central thalamus drive these opposing EEG rhythms, lead to dramatic differences in the spatial extent of forebrain recruitment, and switch the region's downstream influence on cortex from excitation to inhibition. Notably, high-frequency EEG patterns evoked during 40 and 100 Hz stimulation associate with robust activation of frontal cortex, motor cortex, somatosensory cortex, and striatum – regions that receive widespread glutamatergic projections from intralaminar nuclei (*Jones and Leavitt, 1974*; *Berendse and Groenewegen, 1990*; *Groenewegen and Berendse, 1994; Smith et al., 2004; Hoover and Vertes, 2007*; *Hunnicutt et al., 2014*). On the other hand, slow-wave oscillations evoked during 10 Hz stimulation are associated with limited forebrain activation and strong inhibition of somatosensory cortex. The frequency-dependent generation of spindle-like oscillations, which are known to underlie brain synchronization at the onset of sleep (*Steriade et al., 1993*), suggest that these differences may in part be due to the engagement of thalamocortical networks responsible for sleep and loss of perceptual awareness during 10 Hz stimulation. In the context of our findings, it is also noteworthy that for some rat strains with absence seizures, a specific 10 Hz generator in somatosensory cortex has been independently proposed (*van Luijtelaar and Sitnikova, 2006*).

The broadly contrasting cortical responses between low and high frequencies of central thalamus stimulation imply the frequency-dependent activation of a GABAergic population. In this study, we investigated the behavior of ZI – a region rich in GABAergic neurons that also sends direct inhibitory projections to sensory thalamus and sensory cortex (*Lin et al., 1990*; *Benson et al., 1992*; *Nicolelis et al., 1995*; *Kolmac and Mitrofanis, 1999*; *Barthó et al., 2002*). We found that spindle-like oscillations were uniquely evoked when central thalamus was stimulated at 10 Hz and somatosensory cortex was inhibited. Stimulation at this frequency in asleep rats also evoked absence seizure-like freezing and spike-waves in a majority of animals, a cortical pattern known to be modulated by ZI projections (*Shaw et al., 2013*). Importantly, suppressing the incertal activity evoked by 10 Hz central thalamus stimulation with halorhodopsin reduced the cortical inhibition (*Figure 5H*), suggesting a key role for ZI in modulating this response. Indeed, it has been previously suggested that rhythmic incertal activity contributes to membrane hyperpolarizations and sustained high-voltage cortical rhythms through GABAergic incertofugal pathways (*Shaw et al., 2013*). Our data support this hypothesis, and link such a pathway to whole-brain, directly visualized fMRI activity patterns.

Given the presence of GABAergic projections from ZI to central thalamus (*Barthó et al., 2002*), activity in ZI may also act to limit forebrain activation, as observed with ofMRI during 10 Hz

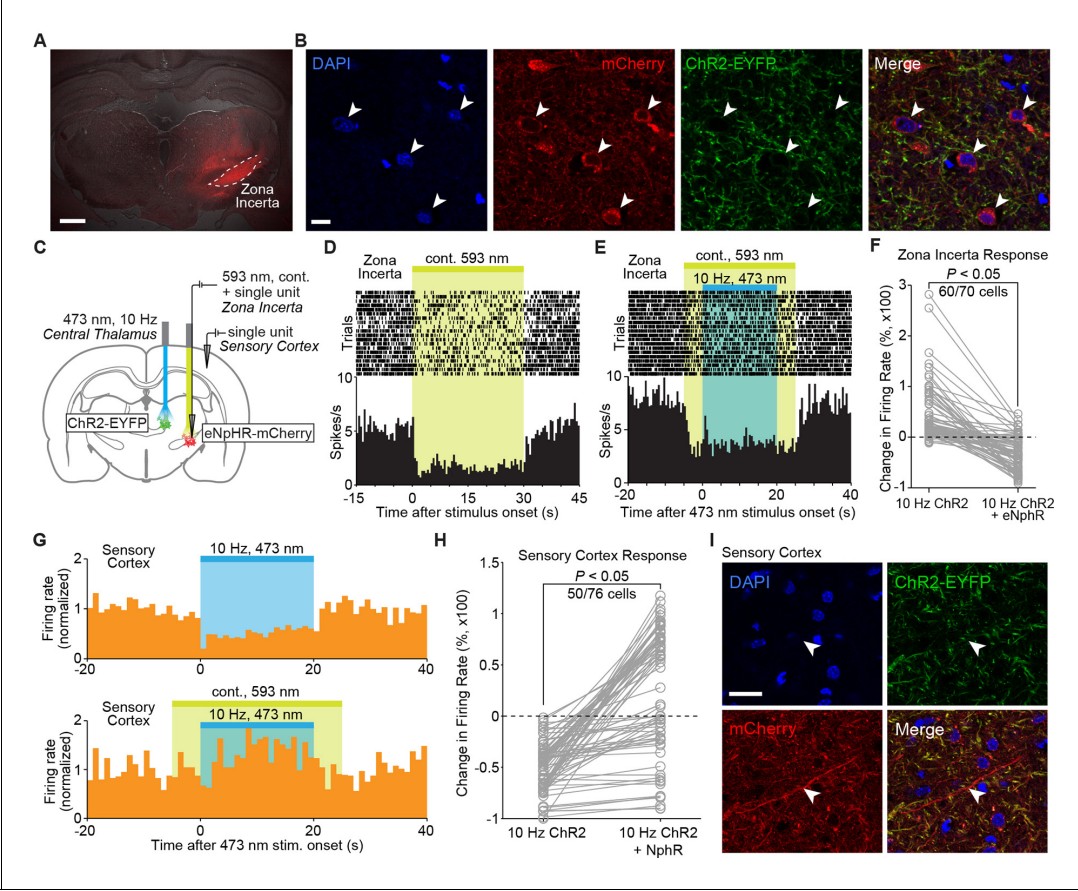

**Figure 5.** Cortical inhibition driven by 10 Hz central thalamus stimulation depends on normal incertal processing. (A) Wide-field fluorescence image shows robust eNpHR-mCherry expression spatially localized to the right zona incerta. Scale bar, 1 mm. (B) Confocal images show eNpHR-mCherry localized to somatic membrane of neurons in zona incerta. Scale bar, 10 μm. Two hundred and nine out of 882 DAPI-stained cells co-expressed mCherry in ZI (24%, n = 2 animals). (C) Schematic of cortical electrophysiology recordings during 10 Hz stimulation of central thalamus and continuous (cont.) inhibition of zona incerta using ChR2 and eNpHR, respectively. (D) Peri-event time histogram of a representative neuron in zona incerta whose firing rate is suppressed during activation of eNpHR with 593 nm light. (E) Peri-event time histogram of a representative neuron in zona incerta whose firing rate remains suppressed throughout the period of 10 Hz central thalamus stimulation during eNpHR activation (compare to *Figure 4C*). (F) Activation of eNpHR in zona incerta significantly reduces the change in incertal firing rate evoked by 10 Hz central thalamus stimulation in 60 of 70 neurons (p < 0.05, one-sided Wilcoxon rank sum test). Changes in firing rate are normalized to pre-stimulation levels. (G) Peri-event time histograms from a representative cortical neuron show that the inhibitory response evoked by 10 Hz central thalamus stimulation is reversed by simultaneously suppressing activity in zona incerta. Firing rates are normalized to the average pre-stimulation values. (H) Quantification of evoked changes in cortical firing rate during 10 Hz central thalamus stimulation with and without concurrent eNpHR activation. 50 out of 76 cells exhibit reduced inhibition when central thalamus stimulation is paired with eNpHR activation (p < 0.05, Wilcoxon rank sum test over 1 s bins). Changes in firing rate are normalized to pre-stimulation levels. (I) Confocal images show mCherry-positive axonal projections from zona incerta in somatosensory cortex. Scale bar, 20 μm. See also *Figure 5—source data 1,2*.

The following source data and figure supplement are available for figure 5:

**Source data 1.** Evoked changes in incertal firing rate during 10 Hz central thalamus stimulation with and without concurrent eNpHR activation.
**Source data 2.** Evoked changes in cortical firing rate during 10 Hz central thalamus stimulation with and without concurrent eNpHR activation in zona incerta.
**Figure supplement 1.** Wide-field fluorescence image of eNpHR expression in zona incerta, overlaid with the estimated cone of activated eNpHR (i.e. inhibited neurons) shown to scale.

stimulation, through incertal-thalamic feedback. This incerto-thalamic pathway would parallel the previously reported gating of ascending sensory information at the level of thalamus by ZI

(*Trageser and Keller, 2004*; *Lavallée et al., 2005*). The hypothesized feedforward and feedback inhibition via ZI both suggest a direct projection from central thalamus to ZI, which our fluorescence imaging data support (*Figure 4H*). However, we note that previous tracing studies failed to identify thalamic input specifically from intralaminar nuclei to ZI (*Shammah-Lagnado et al., 1985*). In summary, our findings provide the first demonstration that arousal regulation driven by central thalamus has a causal and frequency-dependent influence on ZI, and that suppressing the recruitment of ZI modulates the brain-wide dynamics driven by central thalamus stimulation. Specifically, our results suggest that the frequency-dependent depression of cortical activity is in part mediated by extrinsic inhibitory signals originating from ZI.

An additional mechanism that could contribute to the evoked suppression of cortex is feedforward thalamocortical inhibition – the process by which relay neurons drive inhibitory post-synaptic potentials (IPSPs) in pyramidal cells via fast spiking cortical interneurons (*Agmon and Connors, 1991*; *Porter et al., 2001*; *Cruikshank et al., 2007*). Low-frequency (10 Hz) stimulations of certain thalamic nuclei in vivo yield strong hyperpolarization of cortical neurons putatively via this process (*Castro-Alamancos and Connors, 1996a*; *1996c*). It is also interesting to note that the 'recruiting response', characterized by an enhanced cortical response during low frequency electrical stimulation of intralaminar nuclei (*Morison and Dempsey, 1942*), is hypothesized to originate from this inhibition (*Castro-Alamancos and Connors, 1997*). Our observation that individual stimuli sometimes trigger spikes in cortex (*Figure 3—figure supplement 1*) is consistent with the possibility that these phenomena occur during the delivered 10 Hz optogenetic stimuli. However, intracellular and laminar recordings are needed to more conclusively resolve this issue. More recently, a study utilizing optogenetic targeting showed that non-specific 'matrix' thalamocortical neurons preferentially drive inhibitory interneurons in cortical layer I (*Cruikshank et al., 2012*). Moreover, they found that IPSPs generated by stimulation of matrix neurons (which constitute the nuclei targeted here [*Jones, 2001*]) remain sustained during repeated stimuli compared to those evoked by stimulation of non-matrix neurons. Given the above findings, it is possible that interneuron-mediated thalamocortical inhibition, in addition to the demonstrated role of ZI, may contribute to the observed cortical responses. However, to the best of our knowledge, there have been no in vivo studies demonstrating that cortical interneuron-to-pyramidal cell inhibition is stronger at low frequencies of intralaminar thalamic stimulation compared to high frequencies.

In addition to providing novel insight into the mechanisms of arousal regulation by central thalamus, our study offers important insight into the cellular origins of the fMRI BOLD signal. While there is a growing body of evidence suggesting that negative BOLD signals reflect local decreases in neuronal activity (*Shmuel et al., 2006*; *Pasley et al., 2007*; *Allen et al., 2007*; *Devor et al., 2007*; *Sotero and Trujillo-Barreto, 2007*), the nature of this signal remains a subject of debate and holds significant potential for the interpretation of functional imaging studies (*Schridde et al., 2008*; *Ekstrom, 2010*). It has been shown that different sensory stimuli can evoke positive and negative BOLD signals in the same cortical area, which are linked to increases and decreases in neural activity, respectively (*Shmuel et al., 2006*). Building upon these studies, we found that direct stimulation of central thalamus excitatory neurons at different frequencies leads to activation or suppression of neuronal activity in a downstream cortical location, which is coupled with positive and negative BOLD signals, respectively (*Figure 3*). These findings strongly support the hypothesis that a major component of the negative BOLD signal derives from decreases in neuronal activity and are consistent with previous reports of tight neural-hemodynamic coupling in the somatosensory cortex (*Huttunen et al., 2008*).

Our results are also consistent with a previous study by Logothetis et al., which showed that low frequencies (<50 Hz) of electrical microstimulation in the thalamic lateral geniculate nucleus evoke negative BOLD responses in the monosynaptically connected V1 cortex, while higher frequencies (100–200 Hz) evoke positive BOLD responses in the same region (*Logothetis et al., 2010*). We observed similar results in the somatosensory cortex, which is monosynaptically connected to the stimulated intralaminar nuclei (*Van der Werf et al., 2002*) (*Figure 2—figure supplement 1*). The study by Logothetis et al. also found that cortical regions which are polysynaptically connected to the lateral geniculate nucleus, such as V2, even exhibit negative BOLD responses at high frequencies of stimulation (>60 Hz). It was proposed that these polysynaptic deactivations result from frequency-dependent disynaptic inhibition, the process by which pyramidal cells in cortex inhibit local and remote pyramidal cells via GABAergic interneurons. Unlike the study by Logothetis et al., we did not

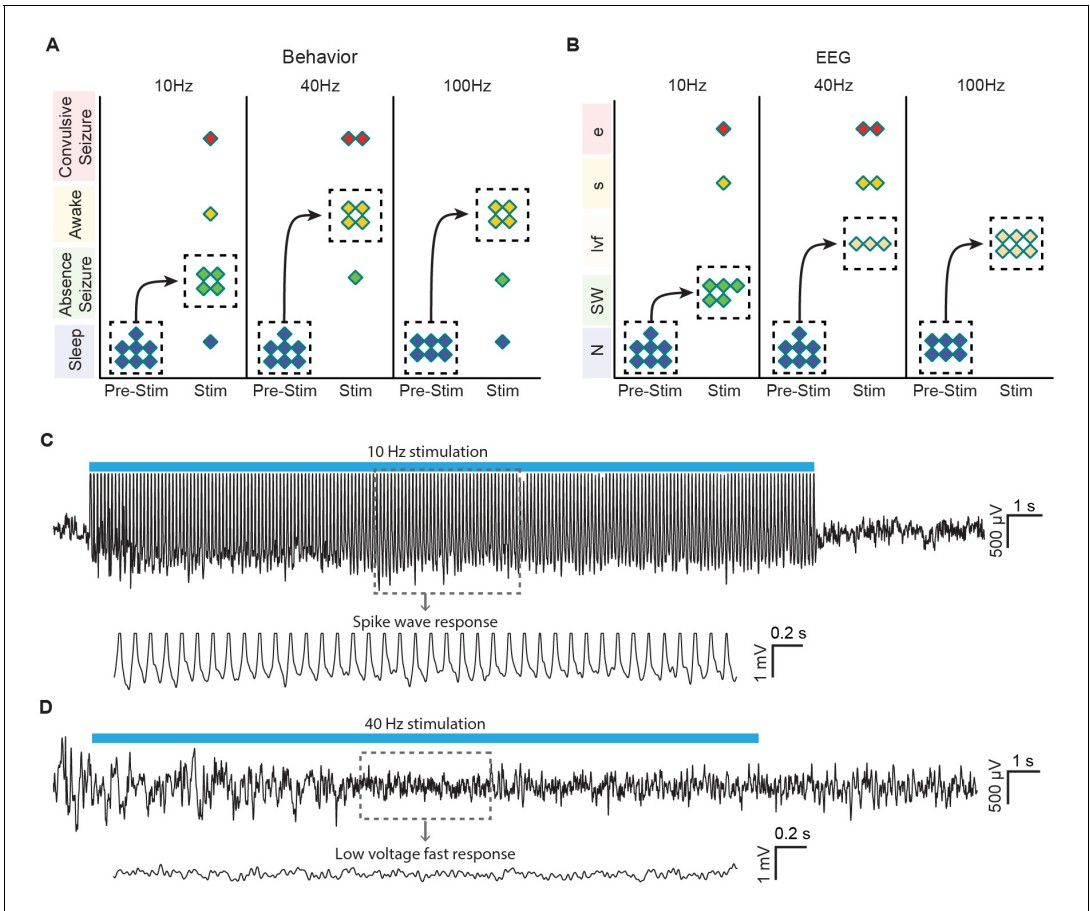

**Figure 6.** Optogenetic stimulation of central thalamus in asleep animals modulates brain state in a frequency-dependent manner. (**A**) Low-frequency stimulation (10 Hz) in a majority of animals (n = 4/7) evokes behavioral absence seizures, while high-frequency stimulations (40 and 100 Hz) cause a majority of animals to awaken (n = 4/7 and 4/6, respectively). Dashed boxes indicate the most common response for each frequency, with arrows indicating the corresponding transition from sleep. (**B**) Low-frequency stimulation typically evokes spike-wave responses in EEG (n = 5/7), consistent with the behavioral reading of absence seizures. The most frequent EEG response during high-frequency stimulations is low voltage fast (n = 3/7 and 6/6), indicative of arousal. N, normal. SW, spike-wave. lvf, low voltage fast. s, spiking. e, evolving seizure. (**C,D**) Representative traces of EEG responses classified as spike-wave and low voltage fast. Insets show 4 s magnification. Importantly, these EEG patterns match those recorded under anesthetized conditions (*Figure 4G*), further linking the responses visualized with ofMRI to the reported behavioral responses. See also *Figure 6—figure supplement 1*. EEG: Electroencephalography.

The following figure supplement is available for figure 6:

**Figure supplement 1.** Pre-stimulus activity is consistent across frequencies of stimulation in asleep rats, as quantified with EEG bandpower in delta, theta, alpha, and beta bands. EEG: Electroencephalography.

observe significant negative BOLD signals in either mono- or polysynaptically connected regions of cortex during high frequencies of stimulation. Furthermore, given the bias of corticocortical disynaptic inhibition toward higher frequencies (*Silberberg and Markram, 2007*), this microcircuit is unlikely to be driving the observed suppression of cortex during 10 Hz central thalamus stimulation.

In the context of electrical stimulation, our study helps dissociate the confounding effects of (a) delivering stimulation at a certain frequency (which can preferentially recruit certain neuronal elements (*McIntyre and Grill, 2002*)) and (b) the excited neuronal population firing at a specific frequency. With electrical stimulation, it has been impossible to dissociate these two effects in vivo, since the frequency of stimulation and preferential recruitment of specific neuronal populations could not be decoupled (*McIntyre and Grill, 2002*). This made it difficult to explain, for example, the relationship between stimulation parameters and the therapeutic efficacy of DBS. Using targeted, temporally precise, optogenetic stimulation in the current study allowed us to selectively

excite a single group of neuronal elements and identify their specific role in creating distinct modes of network function. The use of electrical stimulation instead would have prevented us from gaining this unique insight into the specific role of excitatory central thalamus neurons and their spiking frequency.

Finally, in the context of central thalamus DBS, our study offers important insight into the identification of proper stimulation targets and parameters that are needed before the therapeutic application of central thalamus stimulation can reach its full clinical potential. In particular, the images from ofMRI experiments (*Figure 2* and *Figure 2—figure supplement 3*) reveal dramatic differences in global brain dynamics that can result from controlling one parameter of stimulation (i.e. frequency). Furthermore, the widespread activation of cortex and striatum observed at high frequencies of stimulation adds to a growing body of evidence that the central thalamus is a highly appropriate target for the remediation of acquired cognitive disabilities via forebrain recruitment. In a more general context that extends beyond stimulation of central thalamus, the ofMRI techniques we employ here are generalizable and can be universally applied to study the mechanisms underlying DBS for other target regions and disorders. With this knowledge, stimulation paradigms can be optimized to accelerate clinical translation for a wide range of neurological disorders that currently lack such treatment, paving the way for the development of next-generation DBS therapies.

## Materials and methods

### Animals

Female Sprague-Dawley rats (>11 weeks old, 250-350 g) were used as subjects for all thalamic injections. Animals were individually housed under a 12 hr light–dark cycle and provided with food and water *ad libitum*. Animal husbandry and experimental manipulation were in strict accordance with National Institute of Health, UCLA Institutional Animal Care and Use Committee (IACUC), and Stanford University IACUC guidelines.

### Viral injections and fiber placement

pAAV5-CaMKIIa-hChR2(H134R)-EYFP-WPRE plasmid was obtained from the Deisseroth lab at Stanford University. Concentrated virus was produced at the vector core of the University of North Carolina at Chapel Hill. Rats were anesthetized with isoflurane (induction 5%, maintenance 2–3%; Sigma-Aldrich, St. Louis, MO) and secured in a stereotactic frame. Standard procedures for sterile surgery were followed. Buprenorphine was administered to minimize pain. Artificial tears were applied to the eyes. The head was shaved, and 70% ethanol and betadine were applied to the bare scalp following a midline incision. A small craniotomy was performed with a dental drill above the targeted coordinate. Two microliters of virus were injected through a 34-gauge needle (World Precision Instruments Inc., Sarasota, FL) at 150 nl/min with a micro-syringe pump controller at the desired coordinates in central thalamus or other subcortical targets for control experiments: **I**) CL and PC nuclei of central thalamus (-3.2 mm AP, +1.5 mm ML, -5.6 mm DV; n = 47 animals for imaging); **II**) ventral posteromedial nucleus (-2.5 mm AP, +2.6 mm ML, -6.0 mm DV); **III**) anterior thalamic nuclei (-3.1 mm AP, +1.8 mm ML, -5.3 mm DV); **IV**) posterior thalamic nuclei (-4.6 mm AP, +1.8 mm ML, -5.0 mm DV); **V**) intermediate hippocampus (-5.8 mm AP, +5.2 mm ML, -3.4 mm DV, n = 8 animals). All injections were made in the right hemisphere. The syringe needle was left in place for an additional 10 min before being slowly withdrawn. Custom-designed guide cannulas (Plastics One) or fiber-optic cannulas (Doric Lenses Inc.) were mounted on the skull and secured using metabond (Parkell). Incisions were sutured, and animals were kept on a heating pad until recovery from anesthesia. Buprenorphine was injected subcutaneously twice daily for 48 hr post-operatively to minimize discomfort. The original cohort of 47 central thalamus animals was further refined to a group of 18 after screening for implant locations less than 0.85 mm away from the target coordinate (estimated with T2 MRI scans; *Figure 1B*). Two additional animals were excluded due to lack of thalamic activation, leaving 16 animals for analysis.

In a second cohort of rats, concentrated AAV5-hSyn-eNpHR3.0-mCherry virus produced at the University of North Carolina at Chapel Hill vector core was injected into the right ZI (-3.96 mm AP, +2.8 mm ML, +7.4 mm DV, n = 4 animals) after completion of the ChR2 injection into the central thalamus as described above. Both injections were performed during the same surgery. 0.5 µl of

eNpHR virus were injected through a 34-gauge needle at 100 nl/min. Following the injection, the syringe needle was left in place for approximately 10 min before being slowly withdrawn. Recovery details were the same as those described above.

## ofMRI data acquisition

fMRI scanning was performed using a 7T Bruker Biospec small animal MRI system at UCLA. Animals were initially anesthetized with 5% isoflurane and intubated before placement onto custom-made MRI-compatible cradles with ear and tooth securement. A 39 mm outer diameter, 25 mm inner diameter custom-designed transmit/receive single-loop surface coil was centered over the region of interest on the skull to maximize signal-to-noise ratio. An optical fiber of 62.5 μm core diameter was connected to a 473 nm laser source (Laserglow Technologies, Toronto, Canada) and coupled with the implanted fiber-optic cannula. A single ofMRI scan consisted of a block design with six 20 s pulse trains of light (10, 40, or 100 Hz in randomized order) delivered once per minute over 6 min. Five to six consecutive scans were collected during each session. For all experiments, the optical fiber output power was calibrated to 2.5 mW. A duty cycle of 30% was used across frequencies to maintain the total amount of light delivery, resulting in unique pulse widths of 30, 7.5, and 3 ms for 10, 40, and 100 Hz, respectively. In a series of control experiments using a second cohort of animals with validated probe locations (n = 3), the duty cycle was varied while the pulse width was held constant at 3 ms (*Figure 2—figure supplement 3*).

During fMRI scanning, animals were placed into the iso-center of the magnet while artificially ventilated (45~60 strokes/min) under light anesthesia using a ventilator and calibrated vaporizer with a mixture of $O_2$ (35% ), $N_2O$ (63.5%), and isoflurane (1.3–1.5% ). To ensure stable BOLD signals, expiratory $CO_2$ was kept at 3–4% and body temperature was maintained at 36.5–37.5°C using heated airflow. T2-weighted high-resolution anatomical images were acquired with a fast spin echo sequence prior to fMRI scanning to check for brain damage and validate the optical fiber's location (137 μm resolution in-plane resolution with $35\times35$ mm$^2$ FOV, 0.5 mm slice thickness, 32 coronal slices). Gradient recalled echo (GRE) BOLD methods were used to acquire fMRI images during photostimulation. The fMRI image acquisition was designed to have $35\times35$ mm$^2$ in-plane field of view (FOV) and $0.5\times0.5\times0.5$ mm$^3$ spatial resolution with a sliding window reconstruction to update the image every repetition time (TR) (*Fang and Lee, 2013*). The two-dimensional, multi-slice gradient-echo sequence used a four-interleave spiral readout (*Glover and Lee, 1995*; *Kim et al., 2003*), 30° flip angle, 750 ms TR, and 12 ms echo time, resulting in 23 coronal slices (128 × 128 matrix size). The spiral k-space samples were reconstructed through a two-dimensional gridding reconstruction method (*Jackson et al., 1991*). Finally, real-time motion correction was performed using a previously described GPU-based system (*Fang and Lee, 2013*). Scans with significant motion, identified by careful visual inspection for spiral artifacts and activations at the boundary of the brain, which indicates large motion, were excluded from analysis. This condition for exclusion was established prior to data collection.

## fMRI data analysis

All fMRI data processing was performed using the Matlab software environment (MathWorks, Inc., Natick, MA) and mrVista (Stanford Vision and Imaging Science and Technology Laboratory, Stanford, CA; http://web.stanford.edu/group/vista/cgi-bin/wiki/index.php/MrVista). Motion-corrected images belonging to consecutive scans of the same stimulation paradigm and scanning session were first averaged together. The average 4D images were then aligned to a common coordinate frame, using a six degree-of-freedom rigid body transformation. If multiple scanning sessions were performed on the same animal at the same frequency (typically 1, at most 4), the resulting images from each session were first averaged together before any inter-subject analysis to achieve maximum signal-to-noise ratio while weighting the images from all animal subjects equally.

Time series were calculated for each voxel in these individual-animal images as the percent modulation of the BOLD signal relative to a 30 s baseline period collected prior to stimulation. Boxcar detrending with a window size of 1 min was also performed to correct for possible scanner drift. Next, a coherence value was calculated for each voxel's time series as the magnitude of its Fourier transform at the frequency of repeated stimulation blocks (i.e. 1/60 Hz) divided by the sum-of-squares of all frequency components (*Engel et al., 1997*). Voxels with a coherence value greater

than 0.35 were considered to be significantly synchronized to stimulation. Assuming Gaussian noise and ~470 degrees of freedom (computed using the SPM software environment), the Bonferroni-corrected p value for this threshold can be estimated to be less than $10^{-9}$ (*Bandettini et al., 1993*). Activation volume (*Figure 2*) was defined as the number of significant voxels that exhibited a positive response within a predefined region of interest, multiplied by the volume per voxel. Positive responses were identified as those having a phase in the interval $[0, \pi/2] \bigcup [3\pi/2, 2\pi]$. Phase represents the temporal shift of the response when it is modeled as a sinusoid and was calculated as the argument of each voxel's Fourier transform at the frequency of repeated stimulation blocks (i.e. 1/60 Hz).

Hemodynamic response functions (HRFs) were calculated as the average 60 s response of a voxel's six-cycle, 6-min time series. Time series and HRFs displayed for figures were generated by averaging the mean time series or mean HRF of voxels with a coherence value greater than 0.35 in the corresponding ROI across animals. For *Videos 1–3*, the first data point's value was subtracted from each voxel's HRF to define its relative percent modulation from the onset of stimulation.

To generate average activation maps (*Figure 2*), the 4D fMRI images from experiments at the same stimulation location and frequency were normalized and averaged together across animals. The averaged images were then processed according to the above Fourier domain analyses. Coherence values were overlaid onto all voxels having a coherence above the 0.35 threshold. Warm and cool colormaps generated using Matlab's 'hot' and 'winter' functions were used for positive and negative responses, respectively, to illustrate the localization of negative BOLD to the somatosensory cortex. These activation maps were overlaid onto corresponding T2-weighted anatomical images with a digital standard rat brain atlas (*Paxinos and Watson, 2005*). The same atlas was used to segment ROIs. An identical analysis pipeline was used for activation maps in *Figure 2—figure supplement 3* with a representative animal.

## EEG electrode implantation

EEG electrodes were implanted upon completion of ofMRI experiments in a subset of animals. Surgical preparation and recovery details were the same as those used for virus injection. Stainless steel screws (0–80, 1.5 mm diameter, Plastics One) attached to 2 cm of insulated wire (30 gauge, R30Y0100, Wire Wrapping Wire, O.K. Industries) were used as EEG electrodes and secured on the skull using dental cement. The recording electrode was placed approximately 2 mm anterior of bregma and 2 mm to the right of midline. The reference electrode was located approximately 5 mm anterior of bregma and 3 mm to the left of midline (*Horner et al., 2003*).

## Video-EEG acquisition and analysis

Prior to video-EEG recording, animals were anesthetized under 5% isoflurane for approximately 5 min for optical fiber coupling and EEG wire connection. Animals were then transferred to a light- and sound-controlled experimental chamber where they were allowed to move freely. Behavioral experiments began after animals recovered from anesthesia and subsequently fell asleep for 15 min (as indicated by lack of motion and real-time EEG output readings). For each experiment, the animal was video-recorded during 5 min of sleep, followed by 20 s of optical stimulation (10, 40, or 100 Hz, 473 nm laser, 2.5 mW laser power, 30% duty cycle), and then an additional 5 min post-stimulation period. EEG data was acquired throughout the experiment at 1 kHz with an MP150 data acquisition unit and EEG100C amplifier (Biopac Systems Inc., Santa Barbara, CA), using EL254S Ag-AgCl electrodes and Gel102 conductive EEG paste. A digital camera was used to video-record the experiment. All behavioral experiments were performed during the animals' light cycle.

EEG recordings were classified using the Biopac Acqknowledge software by an experienced electroencephalographer blind to treatment into a single best category: normal, low voltage fast, spikes, spike-waves, or evolving electrographic seizure. Video clips paired to each EEG recording were classified into one of the following categories to further assess the animal's brain state: sleep (i.e. no change), awakening (animal is alert and exploring), absence seizure (animal is immobile and appears frozen for the duration of stimulation, but returns to a sleeping state once stimulation ends), or convulsive seizure. All observed behavioral responses could be classified into one of these categories. Band power in *Figure 6—figure supplement 1* was quantified using Matlab's 'bandpower' function and normalized by the signal's total power from 0 Hz to one half the sampling rate (500 Hz).

## In vivo electrophysiology

Upon completion of ofMRI and EEG behavioral experiments, in vivo electrophysiology experiments were performed in a subset of animals. Animals were anesthetized with 5% isoflurane for induction and maintained at 2–3% until any craniotomies were complete. Isoflurane was kept at 0.8–1.2% during the recording session, and artificial tears were applied to the eyes. Recordings in *Figures 4* and *5* were performed under ventilation conditions identical to fMRI experiments. After securing the animal within a stereotactic frame, small craniotomies were performed using a dental drill above the region of interest. For stimulation, the cannula implanted at central thalamus was connected to a 473 nm laser source (Laserglow Technologies) with an output power level of 2.5 mW via an optical fiber. The cannula implanted at ZI was connected to a 593 nm laser source (Laserglow Technologies) calibrated to 2.5–3.0 mW. An acute 16-channel microelectrode array was targeted to the recording site using stereotactic instruments (NeuroNexus Technologies; A1x16 standard model linear electrode array for local and cortical recordings; V1-16-Poly2 polytrode array for ZI recordings; $0.35 \pm 0.5$ MOhm impedance). A stainless steel reference screw was placed above the cerebellum. Continuous field potential and single unit spiking events were recorded using the Plexon omniplex system with plexcontrol software (Plexon Inc., TX). When only ChR2 was activated, recordings were performed for 20 s without stimulation, followed by repeated stimulation cycles (20 s on, 40 s off) at 10, 40, or 100 Hz with 30% duty cycle. When ChR2 and eNpHR were activated together, the same stimulation paradigm was followed, except that a 30 s period of continuous 593 nm light delivery began 5 s before the onset of ChR2 excitation. When only eNpHR was activated (*Figure 5D*), a 20 or 30 s period of continuous 593 nm light delivery was used, with 40 or 30 s periods of no light delivery between repeated cycles, respectively. For single unit responses, the Plexon multichannel acquisition processor was used to amplify and band-pass filter the neuronal signals (150 Hz – 8 kHz). Signals were digitized at 40 kHz and processed to extract action potentials in real-time. To separate the field potential, we used a low-pass filter (200 Hz cutoff frequency, 4-pole Bessel filter) and downsampled signals to 1 kHz. Simultaneous EEG data was collected at 1 kHz during ZI recordings in *Figure 4* using the MP150 data acquisition unit and EEG100C amplifier (Biopac).

## ZI electrophysiology analysis

For the analysis in *Figure 4*, field potential recordings were high pass filtered with a cutoff frequency of 2 Hz to eliminate respiratory artifacts. Spindle-like oscillations (SLOs) occurring during the stimulus were then quantified on a per trial basis using a post-hoc custom algorithm (see *Source code 1*). Briefly, an SLO was identified when the recording's magnitude reached at least 6 standard deviations above its mean absolute value. If the recording did not exceed this value for the preceding 500 ms, and was above this value for at least 2% of samples over the next 500 ms, an SLO was counted. This method of quantification accurately captured the large-amplitude oscillations that could be visually discerned (see *Figure 4D*).

## Fluorescence imaging and immunohistochemistry

Upon completion of in vivo of MRI, behavioral, and electrophysiology experiments, rats were deeply anesthetized with isoflurane in a knockdown box and transcardially perfused with 0.1M phosphate-buffered saline (PBS) and ice-cold 4% paraformaldehyde (PFA) in PBS. Brains were extracted and fixed in 4% PFA overnight at 4°C. The brains were equilibrated in 10%, 20%, and then 30% sucrose in PBS at 4°C. Coronal sections (50 μm) were prepared on a freezing microtome (HM 430 Sliding Microtome, Thermo Scientific Inc.). Consecutive sections (500 μm apart) were mounted and examined with a fluorescence microscope (Leica EL6000). For quantitative immunohistochemistry (*Figure 1—figure supplement 1*), free-floating sections were processed with 5% normal donkey serum, and 0.4% Triton X-100 for 60 min. Sections were then exposed at 4°C for 48 hr to primary antibodies against mouse monoclonal CaMKIIα (CaMKIIα, 1:500, 05–532, Millipore, Billerica, MA). After washing with PBS, sections incubated for 2 hr at room temperature with Alexa Fluor 647-conjugated AffiniPure donkey anti-mouse IgG (1:250, Jackson Laboratories, West Grove, PA). Slices were then washed and mounted (DAPI-Fluoromount G, SouthernBiotech, Birmingham, AL). Immuno-fluorescence was assessed with a laser confocal microscope (Leica CTR 6500).

For high-resolution, whole-brain fluorescence imaging (*Figures 1A*, *4H*, and *Figure 2—figure supplement 1*), frozen brains were embedded using stainless steel Tissue-Tek base molds and Neg-

50 embedding medium (Richard-Allan Scientific [Thermo]; n = 2 animals) (*Pinskiy et al., 2013*). Post-freezing, the Neg-50 embedded brain was sectioned on a Microm HM550 cryostat using the tape-transfer method with all sections mounted directly onto slides. Alternating sections, cut at 20 µm, were separated to form two distinct series per brain. One slide series of the sectioned material was processed for Nissl cell body staining, using a thionin-based protocol and coverslipped with DPX mounting medium. The alternate series was dehydrated and directly coverslipped with DPX for fluorescence imaging. Whole-slide digital imaging was performed using a Hamamatsu NanoZoomer 2.0-HT system at 0.46 µm/pixel, with fluorescence scans at 12-bit depth using a tri-pass filter cube. Following data conversion to lossless jp2 (JPEG2000), individual brain sections were aligned and registered using rigid 2-D image transformation.

## Statistics

All statistical tests were performed in Matlab. Non-parametric tests were used throughout the analysis. For in vivo electrophysiology measurements at thalamus and ZI, one-tailed Wilcoxon signed-rank tests were used to evaluate changes in firing rate between the pre-stimulation and stimulation periods. For measurements at sensory cortex in *Figure 3*, a two-tailed version of the test was used to evaluate either increases or decreases in firing rate. For results in *Table 1*, the average pre-stimulus firing rate (20 s bin) was compared to the average firing rate of four 5 s bins over the 20 s period of stimulation using a one-tailed Wilcoxon signed rank test, uncorrected for multiple comparisons. One-sided Wilcoxon rank sum tests were used to evaluate differences in SLO occurrence (*Figure 4E*), as well as changes in cortical or incertal firing when eNpHR activation was coupled with central thalamus stimulation (*Figure 5F,H*). For electrophysiology results, independence was assumed between repeated trials. All other assumptions for these tests were satisfied. For volumetric comparisons in *Figure 2,* one-sided Wilcoxon signed-rank tests were used to identify increases in the volume of BOLD activation between 10 and 40 Hz and 10 and 100 Hz (corrected for multiple comparisons). Note that variance was generally similar across groups being compared. Significance was determined at the α = 0.05 cutoff level. No statistical methods were used to estimate sample size. All statistical tests used to compare changes with frequency (*Figure 2* and *Figure 2—figure supplement 2B*) were performed pairwise, with an equal number of animals used for each frequency.

## Acknowledgements

JHL would like to acknowledge Karl Deisseroth for providing the DNA plasmids. The authors thank the Lee Lab members for their contribution to ofMRI experiments.

## Additional information

### Funding

| Funder | Grant reference number | Author |
|---|---|---|
| National Science Foundation | CAREER Award, 1056008 | Zhongnan Fang<br>Jin Hyung Lee |
| Okawa Foundation for Information and Telecommunications | Research Grant Award | Jin Hyung Lee |
| Alfred P. Sloan Foundation | Sloan Research Fellowship | Jin Hyung Lee |
| National Institutes of Health | NIH Director's New Innovator Award | Jia Liu<br>Hyun Joo Lee<br>Peter Lin<br>ManKin Choy<br>Jin Hyung Lee |
| National Institute of Neurological Disorders and Stroke | R01NS087159 | Hyun Joo Lee<br>Andrew J Weitz<br>Peter Lin<br>ManKin Choy<br>Nicholas Schiff |

| | | Jin Hyung Lee |
| --- | --- | --- |
| National Institute of Biomedical Imaging and Bioengineering | R00EB008738 | Jia Liu<br>Zhongnan Fang<br>Jin Hyung Lee |
| National Institutes of Health | Transformative R01, R01MH087988 | Vadim Pinskiy<br>Alexander Tolpygo<br>Partha Mitra |
| Mathers Charitable Foundation | | Vadim Pinskiy<br>Alexander Tolpygo<br>Partha Mitra |
| Stanford Bio-X | Bioengineering Graduate Fellowship | Andrew J Weitz |
| James and Carrie Anderson Fund for Epilepsy Research | | Robert Fisher |
| Susan Horngren and Littlefield Funds | | Robert Fisher |

The funders had no role in study design, data collection and interpretation, or the decision to submit the work for publication.

## Author contributions

JL, Collected the fMRI data, Acquisition of data; HJL, Performed EEG, in vivo single-unit, and field potential electrophysiology recordings; AJW, Analyzed the fMRI and electrophysiology data and wrote the paper; ZF, Analyzed the data and helped make figures and videos; PL, Conducted behavioral experiments; MKC, Performed quantitative immunohistochemistry; RF, Scored the video-EEG data; VP, AT, Conducted optical brain slice imaging; PM, Advised VP and AT, and helped design the experiments; NS, Helped with the design of the experiments, interpretation of the data, and writing of the paper; JHL, Designed and planned all experiments, supervised and organized the study, analyzed and interpreted data, wrote the paper, and outlined the figures and videos

## Ethics

Animal experimentation: This study was performed in strict accordance with the recommendations in the Guide for the Care and Use of Laboratory Animals of the National Institutes of Health. All of the animals were handled according to approved institutional animal care and use committee (IACUC) protocols from the University of California, Los Angeles (#2010-026) or Stanford University (#20047). All surgeries were performed under isoflurane anesthesia, and every effort was made to minimize suffering.

## Additional files

### Supplementary files

• Source code 1. Matlab script (countSLOs.m) for identification and quantification of spindle-like oscillations.

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
