## [Decision Letter]

[Editors’ note: this article was originally rejected after discussions between the reviewers, but the authors were invited to resubmit after an appeal against the decision.]

Thank you for choosing to send your work entitled "Frequency-selective control of forebrain networks by central thalamus" for consideration at *eLife*. Your full submission has been evaluated by Timothy Behrens (Senior Editor), Andrew King (Reviewing Editor) and three peer reviewers, and the decision was reached after discussions between the reviewers. Based on our discussions and the individual reviews below, we regret to inform you that your work will not be considered further for publication in *eLife*.

Although we were impressed with the interdisciplinary approach used in this study, particularly the combination of optical stimulation and fMRI, as well as its potential therapeutic implications, a number of significant concerns were raised by the reviewers. While some of these can be dealt with by changes to the text, other methodological concerns will likely require further experiments. In particular, the effects of varying the frequency of optical stimulation appear to be confounded by changes in pulse duration and number. Two of the reviewers also felt that the section on the zona incerta was particularly weak.

Reviewer #1:

Liu and colleagues have used a combination of fMRI, optogenetics, electrophysiology and video EEG monitoring in a rodent model to map thalamocortical and forebrain neural circuits. This work has several important contributions. The authors show that mode of stimulation can change response patterns of neurons dramatically in these central circuits. In one case, high-frequency stimulation of excitatory relay neurons in the central thalamus caused a widespread positive BOLD signal in multiple forebrain regions.

In contrast, low frequency (10 Hz) stimulation led to less neural circuit modulation, negative BOLD signal in the somatosensory cortex and loss of consciousness. This low frequency stimulation was also associated with spindle like activity in the zona incerta. The authors postulate that the zona incerta may be playing a roll in the inhibition and spindling through a feedback circuit with the central thalamus.

The authors' work suggests that negative BOLD signals could be associated with large scale inhibition in brain networks.

The work draws a compelling distinction between stimulating at a given frequency and observing population spiking activity at a given frequency after stimulation. The optogenetics disambiguate these two factors by allowing the stimulation of a specific neuronal population.

Finally, the work has enormous translational potential. The authors demonstrate substantial forebrain network activation with stimulation of the central thalamus. This observation clearly speaks to the potential use of these findings to guide developmental work on DBS for therapeutic purposes. In particular, the authors show that changing frequency of stimulation can change the brain response from broad activation of forebrain circuits for high frequency stimulation to decreased arousal and negative BOLD signals for low frequency stimulation. One could imagine using the first stimulation pattern or patterns similar to it help enhance arousal in patients who are minimally conscious or in a vegetative states, whereas the second pattern might prove useful for treating seizures. At any rate, the results offer a principled way to investigate the basic science of DBS and to learn more about the brain's arousal circuits by combining two modalities.

The manuscript is well-written. It is a significant basic neuroscience and translational neuroscience contribution. I have no major concerns.

Reviewer #2:

In this study, Liu et al carry out a multilevel exploration of the role of central thalamus activation in the control of brain states. By using optogenetic activation of the central thalamus in combination with fMRI (and other techniques), they demonstrate how the activation of somas in the central thalamus is enough to induce large changes in cortical BOLD signal, later correlated with single unit firing. The authors demonstrate that while high (40-100 Hz) frequency stimulation of the central thalamus activates different cortical (and subcortical) areas causing arousal, low (10 Hz) frequencies reduce cortical activity. The responses of zona incerta to the same stimulation patterns are compatible with a potential role of this area in the switch between both effects, at least those concerning some areas of the cortex. Even when a definitive causal and mechanistic relationship is not established, the study is relevant for both basic neuroscience – it brings together the single neuron level to the whole brain activation and it gives a feasible systems level interpretation of the phenomena – and also for the applied, clinical implications.

Main concerns:

1) The inclusion of the zona incerta as a potential intermediary of the cortical inhibition at 10 Hz provides a potential mechanism and it is valuable. However the study of this is not well justified in the text. It is not even mentioned in the Introduction and it is not well justified in the Abstract. Why the zona incerta? What inputs does it receive? Is central thalamus projecting to zona incerta? For example in the Introduction all these explanations are given for the pallidum, but then the zona incerta is the one included in the study. In the anatomical study reported in the manuscript, the ZI is never mentioned as a place of distribution of the EYFP axons.

2) In the subsection “Low-frequency central thalamus stimulation drives incertal oscillations” it is said that ZI sends direct gabaergic projections to sensory cortex (somatosensory; Nicolelis, Chaplin and Lin, 1995), and the study is carried out by recording from S1 and Z1. However, this does not explain how the brain state switch would be in prefrontal cortex or other cortical areas. It is possible that the authors have considered this and therefore it should be explicitly discussed in the text.

3) In the study of ZI the 10 Hz thalamic stimulation evokes spike and wave discharges in the EEG but this is not reported in the other recordings in the cortex during 10 Hz stim. 10 Hz CT stimulation is reported silencing the cortex, however spike and wave is a rather visible activity that does not silence the neurons. Shouldn't the spike and wave be seen in the single unit recordings?

4) Subsection “Stimulation of central thalamus relay neurons drives widespread forebrain activation in vivo”, fourth paragraph: stimulating at different frequencies always has the problem of giving the same number of pulses versus giving the stimulation for the same period of time. I understand that here the second option was taken. This means that the number of pulses was (in 6 minutes) 1200 when at 10 Hz, 4800 when at 40 Hz and 12000 at 100 Hz. In order to rule out the effect of the differential number of pulses, and since there are six blocks of repetitions, it would strengthen the conclusions to demonstrate that the differences in cortical activation remain for equivalent number of pulses. Probably the authors already have some data showing this that could be included in a supplemental figure. It is also confusing that Figure 4 shows stimulation periods that are variable and therefore (probably) adjusted to number of pulses. Was a different protocol used in this case?

5) In general there is a lack of description of the control (pre-stimulus) activity. Is the departing point slow wave activity? Is the decrease in firing during 10 Hz stimulation independent from the pre-existing activity patterns?

Reviewer #3:

This study deals with an interesting issue, the role of central thalamic nuclei in controlling cortical activation. However, there are several problems with this paper.

First, the authors seem to have a poor understanding of the effects that thalamic stimulation delivered at different frequencies has on the cortex. This has been studied previously but there is very little reflection of that knowledge here. I can think of a series of studies in the rat that investigated the cortical effects of frequency-dependent electrical stimulation in the VL/VA thalamic region (adjacent to the targeted CL/PC region) in rats. Those studies showed that stimulation in various thalamic nuclei evoke short-latency frequency-depend excitatory cortical responses that can augment (spreading significantly when stimulating in VL/VA) or depress (when stimulating in VPM/VPL). Importantly, at 10 Hz, stimulation always drives strong long-lasting feedforward inhibition, which hyperpolarizes cortical cells immediately after each short-latency excitatory response (e.g. see Fig. 1 in Castro-Alamancos and Connors, 1996b). This inhibition of cortical cells may explain the negative BOLD observed in S1 during 10 Hz stimulation (20 sec); under this stimulation regimen, cortical cells will be hyperpolarized most of the time.

Second, the idea that the suppression of neural activity observed at 10 Hz would be caused by the zona incerta seems unlikely. There are numerous other possibilities. For example, the very robust drive of feedforward inhibition in neocortex at 10 Hz is more parsimonious. In general, this part of the study is very unconvincing.

Third, there are significant methodological issues to consider. Most important is the fact that the authors chose to keep the duty cycle constant across stimulation frequencies. This means that at 10 Hz the light stimulus per cycle lasts 30 ms and at 100 Hz it lasts 3 ms. This is very problematic because the changes that are being attributed to frequency can easily be attributed to pulse duration, which has very significant effects on the responses evoked. Thus, everywhere in the paper where the authors allude to frequency, they are also changing pulse duration, which is a major confound. The only reasonable way to keep the total light duration constant when studying frequency is by increasing the train duration for the lower frequencies and keeping the pulse duration constant.

Fourth, the selectivity of the stimulation, which is based mostly on the placement of the fiber, is a major limitation because the nuclei that are targeted are very small and have complex shapes. It seems inevitable that adjacent structures will also be driven by the light. A Cre-mouse that would limit the expression to the targeted nuclei would be a much better choice in this particular case.

Fifth, regarding the idea that the thalamus serves to drive cortical activation, previous work has shown that neuromodulation of neural activity in sensory thalamus leads to activation or deactivation in the related sensory cortex (Hirata and Castro-Alamancos, 2010). This work is directly related to the study at hand but was not considered here.

[Editors’ note: what now follows is the decision letter after the authors submitted for further consideration.]

Thank you for submitting your work entitled "Frequency-selective control of forebrain networks by central thalamus" for consideration by *eLife*. Your article has been reviewed by two peer reviewers, and the evaluation has been overseen by Tim Behrens (Senior Editor) and Andrew King (Reviewing Editor).

The reviewers have discussed the reviews with one another and the Reviewing Editor has drafted this decision to help you prepare a revised submission.

Several of their original concerns have been satisfactorily addressed in the revised version and the reviewers have acknowledged the improvements made to the manuscript, particularly through the inclusion of the new pulse width and train duration imaging data. Nevertheless, some substantive issues remain that need to be addressed before a final decision can be made (references cited are given below).

1) One of the reviewers argues that your claim that the ZI is the source of cortical inhibition during 10 Hz stimulation is still not well supported. Whilst agreeing that thalamocortical responses evoked in the sensory cortices by stimulating the sensory thalamus at 10 Hz are depressed both in vivo (Castro-Alamancos and Connors, 1996b) and in slices (Cruikshank et al., 2010), the reviewer has pointed out that thalamocortical responses evoked in neocortex by stimulating other thalamic nuclei (midline, intralaminar, and VA/VL) are different in most respects, both in vivo (Castro-Alamancos and Connors, 1996b) and in slices (Cruikshank et al., 2012). Stimulation of these other thalamic nuclei in vivo at 10 Hz produces facilitating (spindle-like) responses in cortex, not depressing responses. In fact, work in slices also supports this. A direct quote from the in vitro paper (Cruikshank et al., 2012) cited by the authors in the rebuttal emphasizes this, "responses during repetitive stimulation were far more sustained for matrix than for core sensory TC pathways".

Since the 1940s it has been known that stimulating thalamic nuclei at different frequencies evokes three main types of responses in cortex (Dempsey and Morison): primary responses, augmenting responses and recruiting responses. A more recent account of this literature, including more recent studies done in the rat (Dempsey and Morison worked in cats), is presented in Castro-Alamancos and Connors, 1997 (not cited by the authors, but highly relevant). Primary responses occur when specific thalamic nuclei are stimulated, such as sensory thalamus. Primary responses depress with frequency. Augmenting and recruiting responses occur when VL, VA, VM, midline or intralaminar thalamic nuclei are stimulated. These responses are similar in that they produce enhanced cortical responses at stimulation frequencies between 7-14 Hz. They differ in their cortical depth profile, indicating that they target different layers (as expected from the known projections), and in the size of the primary response they evoke. VL stimulation evokes augmenting responses with a small primary response (small compared to sensory pathways) in motor cortical areas, which is surface positive. In contrast, recruiting responses are mostly surface negative, lacking the surface positive primary response of augmenting responses. The present study did not record field potential responses in cortex, so it is difficult to know if the responses they evoke at 10 Hz in cortex are augmenting or recruiting. But a consistent finding from intracellular recordings is that strong feedforward inhibition is evoked in cortex in vivo by stimulating the thalamus at 10 Hz (Castro-Alamancos and Connors, 1996c; Castro-Alamancos and Connors, 1996a). In fact, the area stimulated by the authors is known to evoke recruiting responses, which should yield feedforward inhibitory responses with weaker short-latency excitation (surface positivity) than augmenting responses.

The reviewer concludes that by citing papers based on specific thalamic nuclei, such as core sensory thalamic nuclei (Cruikshank et al., 2010), is not appropriate, because it is well known that the pathways stimulated in this study have different response properties.

2) The reviewer has also criticized your argument that high frequency activity evokes increased levels of interneuron-driven cortical inhibition relative to excitation, whereas low frequency activity does not, on the grounds that the relevance of the cited paper (Galarreta and Hestrin, 1998) to the issue at hand is doubtful because this study only deals with inhibition driven by cortical stimuli, not thalamic stimuli. You need to look at feedforward inhibition driven by thalamic stimulation, preferably in vivo. The other cited slice paper (Cruikshank et al., 2012), looked at feedforward inhibition driven by thalamic inputs but only marginally (only 1 example panel in Figure 6 and no group data). It is hard to derive any strong conclusions other than to say that inhibition is there, albeit depressed compared to the 1st stimulus.

3) The reviewer also points out that white matter stimulation does not mimic thalamic stimulation, let alone central thalamus stimulation, so this should not be used as an argument for the effects of thalamic stimulation on cortical activity.

4) In response to your comment that, in contrast to Fig. 1 in Castro-Alamancos and Connors (1996b), you did not observe spikes on every stimulus presentation at 10 Hz, the reviewer points out that this is based on VL stimulation and recording in the motor cortex, which evokes an augmenting response. If in that figure (or Fig. 3 of the same paper) the surface positive field potential component and associated corresponding excitatory drive (which produces the spikes) are eliminated, what is left is a recruiting response, as defined originally by Dempsey and Morison. Similar to spindles, recruiting responses trigger spikes only on some of the waves at 10 Hz. Thus, 3.6 Hz would be consistent with this. It would be helpful if you were to present PSTHs using each stimulus of the 10 Hz train as the marker (Figure 3).

5) Your evidence that inhibition of the zona incerta during 10 Hz central thalamus stimulation reverses the cortical inhibition effect is based on AAV delivery of halorhodopsin, which will not discriminate between inhibitory and excitatory neurons. Consequently, light delivery might not only inhibit ZI cells, but other cells too, including adjacent thalamocortical cells. It is therefore possible that the effects observed are due to inhibition of thalamocortical cells. Similarly, could the mCherry fibers observed in cortex actually be thalamocortical fibers? It is also possible that your stimulation could be activating the NRt, which would drive the observed spindles and the recruiting responses in the cortex at 10 Hz, in which case there would be no need for the extrinsic inhibition purportedly provided by ZI. These possibilities should be addressed in the paper. Unless they can be ruled out, conclusions about the role of the zona incerta should be modified accordingly.

6) In discussing alternative hypotheses of the mechanism of cortical inhibition, including disynaptic thalamocortical inhibition (see Discussion, paragraph four) and disynaptic corticocortical inhibition (see Discussion, paragraph six), you should include relevant literature (e.g. rat in vivo thalamic stimulation), instead of marginally related work (slice studies that used white-matter or cortical stimulation, hippocampal studies, etc.). Care should also be taken to cite the literature accurately. For example, you argue that "central thalamus at different frequencies can lead to distinct behavioral responses" and cite Morison and Dempsey (1942), but Dempsey and Morison did not study behavior. Galarreta and Hestrin (1998) did not study thalamocortical responses (please correct this in the subsection “Central thalamus stimulation frequency controls cortical excitation/inhibition balance”).

References cited:

Cruikshank, S.J., H. Urabe, A.V. Nurmikko, and B.W. Connors, Pathway-specific feedforward circuits between thalamus and neocortex revealed by selective optical stimulation of axons. Neuron, 2010. 65(2): p. 230-45.

Galarreta, M. and S. Hestrin, Frequency-dependent synaptic depression and the balance of excitation and inhibition in the neocortex. Nat Neurosci, 1998. 1(8): p. 587-94.

---

## [Author Response]

[Editors’ note: the author responses to the first round of peer review follow.]

We were very pleased to see that two of the three reviewers found our manuscript suitable for publication, with such comments as “This work has several important contributions,” “The work has enormous translational potential,” and “The study is relevant for both basic neuroscience… and also for the applied, clinical implications.”

We have fully addressed the few criticisms offered by the reviewers through additional clarification in writing, experiments, and analyses in the attached revised manuscript. Below, we provide a point-by-point response to the comments by Reviewers #2 and #3. In the instances where we omitted citing key literature, we sincerely apologize for the oversight and have addressed the issue carefully.

Given the overwhelmingly positive reviews from two of the three reviewers, and our direct rectification of the two issues noted in our rejection letter (frequency vs. pulse width effects, and interpretation of zona incerta recordings) with additional experiments, we request the opportunity to have this revised manuscript re-evaluated by the reviewers.

Reviewer #2:

*1) The inclusion of the zona incerta as a potential intermediary of the cortical inhibition at 10 Hz provides a potential mechanism and it is valuable. However the study of this is not well justified in the text. It is not even mentioned in the Introduction and it is not well justified in the Abstract. Why the zona incerta? What inputs does it receive? Is central thalamus projecting to zona incerta? For example in the Introduction all these explanations are given for the pallidum, but then the zona incerta is the one included in the study. In the anatomical study reported in the manuscript, the ZI is never mentioned as a place of distribution of the EYFP axons.*

We thank the reviewer for their critical feedback of our writing and have revised the manuscript accordingly to address the listed concerns. Specifically, in the Introduction, we provide context for our investigation of zona incerta, citing prior demonstrations of its influence on attention and arousal, and explain that its possible involvement in central thalamus arousal circuits has not yet been studied. Furthermore, in the Abstract, we more explicitly justify the investigation as one to explore the mechanism underlying the negative BOLD signal observed in cortex.

While we speculate that a monosynaptic connection may exist between the transfected neurons in central thalamus and zona incerta based on observed distributions of EYFP axons (see subsection “Cortical inhibition driven by central thalamus stimulation depends on evoked incertal activity”), previous tracing studies have failed to identify significant input to zona incerta from intralaminar nuclei. This clarification has been added to the Discussion. We have kept our discussion of thalamo-striatal projections in the Introduction to provide context for the strong BOLD responses evoked in this region.

*2) In the subsection “Low-frequency central thalamus stimulation drives incertal oscillations” it is said that ZI sends direct gabaergic projections to sensory cortex (somatosensory; Nicolelis, Chaplin and Lin, 1995), and the study is carried out by recording from S1 and Z1. However, this does not explain how the brain state switch would be in prefrontal cortex or other cortical areas. It is possible that the authors have considered this and therefore it should be explicitly discussed in the text.*

To address this concern, we have included several hypotheses in the revised manuscript. For example, during 10 Hz stimulation, cortical deactivation in EEG and the absence of significant forebrain recruitment with fMRI may be controlled by spindle-like oscillations, which were observed in zona incerta, and associated sleep networks (see Discussion). GABAergic projections from zona incerta to central thalamus may also act as a form of negative feedback to suppress the region’s broad thalamocortical projections.

3) In the study of ZI the 10 Hz thalamic stimulation evokes spike and wave discharges in the EEG but this is not reported in the other recordings in the cortex during 10 Hz stim. 10 Hz CT stimulation is reported silencing the cortex, however spike and wave is a rather visible activity that does not silence the neurons. Shouldn't the spike and wave be seen in the single unit recordings?

As correctly summarized by the reviewer, we found that 10 Hz stimulation of central thalamus evokes spike-and-wave discharges in EEG and neuronal silencing in single-unit recordings. Although the reviewer suggests that these phenomena are contradictory, the spatial localization of the two recordings are quite distinct. The decrease in single-unit firing was measured specifically in somatosensory cortex, which was the only region to exhibit a negative BOLD signal (For example, motor and cingulate cortex exhibited positive responses; Figure 2). In contrast, the EEG electrode was placed on the dura and located above frontal cortex. Therefore, the recorded signal represents the integration of a larger volume that likely includes excited regions (e.g. motor and cingulate cortex). Indeed, the reviewer’s comment highlights the novelty of our results. While our observation of cortical spike-and-wave activity is consistent with previous studies of thalamic stimulation, the whole-brain visualization afforded by fMRI allowed us to predict the presence of neuronal silencing in a highly localized region of cortex, which we later went on to confirm with electrophysiology. This demonstrates that the whole-brain visualization afforded by ofMRI is a key advantage over other methods in deconstructing complex neural circuits.

*4) Subsection “Stimulation of central thalamus relay neurons drives widespread forebrain activation in vivo”, Fourth paragraph: stimulating at different frequencies always has the problem of giving the same number of pulses versus giving the stimulation for the same period of time. I understand that here the second option was taken. This means that the number of pulses was (in 6 minutes) 1200 when at 10 Hz, 4800 when at 40 Hz and 12000 at 100 Hz. In order to rule out the effect of the differential number of pulses, and since there are six blocks of repetitions, it would strengthen the conclusions to demonstrate that the differences in cortical activation remain for equivalent number of pulses. Probably the authors already have some data showing this that could be included in a supplemental figure. It is also confusing that Figure 4 shows stimulation periods that are variable and therefore (probably) adjusted to number of pulses. Was a different protocol used in this case?*

The reviewer is correct in their observation that stimulation frequency cannot be varied independently of all other parameters, including pulse width, total light delivery, and pulse train duration. In this experiment, we kept the duty cycle constant in order to mitigate the possible confounding factor of variations in total light delivery, which can lead to heating artifacts (Christie et al., 2013). We apologize if this was not clear to the reviewer, and have clarified this decision in the revised manuscript (Results; Figure 4 legend).

As a result of keeping the duty cycle constant, the pulse width associated with each frequency was necessarily different (30 ms for 10 Hz, 7.5 ms for 40 Hz, and 3 ms for 100 Hz). The alternative option – keeping the pulse width fixed and varying the pulse train duration to maintain total light delivery – would lead to uninterpretable results due to differences in neuronal adaptation that occur over the course of a 20 s pulse train compared to a 2 s pulse train (Note that a 10-fold difference would be required for comparing 10 and 100 Hz).

To address this concern as best as possible, we have performed additional fMRI experiments keeping both the pulse width and train duration constant. Our results show that both the increase in forebrain activation with frequency and the transition from negative to positive BOLD signals in cortex is preserved (Figure 2—figure supplement 3). Although these stimulus trains deliver different amounts of total light, the fact that we see these phenomena while varying frequency and keeping either the total amount of light delivery (Figure 2 and Figure 3) or pulse width (Figure 2—figure supplement 3) constant suggests that frequency is the main factor in driving these effects.

*5) In general there is a lack of description of the control (pre-stimulus) activity. Is the departing point slow wave activity? Is the decrease in firing during 10 Hz stimulation independent from the pre-existing activity patterns?*

To address this concern, we have added a supplementary figure, which shows that the pre-stimulus forebrain EEG activity was consistent across frequencies of stimulation (Figure 6—figure supplement 1). Thus, the differences in evoked behavior and electrographic responses can be attributed to the stimulation paradigm itself.

With regards to the decrease in cortical firing, the magnitude of change during 10 Hz stimulation was well correlated with the pre-stimulus firing rate (n = 46 neurons; R^2^ = 0.78, see Figure 7). This correlation is consistent with expectations, since cells which have a high baseline firing rate can be suppressed more than those with a low baseline firing rate. The data also show that cortical cells exhibiting a wide range of baseline firing rate were inhibited, emphasizing the robustness of central thalamus-driven cortical inhibition. To generate this plot, the average firing rate over the 20 s period before stimulation was compared with the average firing rate during the 5-10 s interval after stimulation began (that is, when the most neurons were silenced; see Table 1).

Author response image 1.**DOI:**
http://dx.doi.org/10.7554/eLife.09215.028

Because the single-unit recordings in cortex were collected without simultaneous forebrain EEG, we cannot compare the magnitude of neuronal silencing to the broader state of cortex prior to stimulation.

Reviewer #3:

*This study deals with an interesting issue, the role of central thalamic nuclei in controlling cortical activation. However, there are several problems with this paper. First, the authors seem to have a poor understanding of the effects that thalamic stimulation delivered at different frequencies has on the cortex. This has been studied previously but there is very little reflection of that knowledge here. I can think of a series of studies in the rat that investigated the cortical effects of frequency-dependent electrical stimulation in the VL/VA thalamic region (adjacent to the targeted CL/PC region) in rats. Those studies showed that stimulation in various thalamic nuclei evoke short-latency frequency-depend excitatory cortical responses that can augment (spreading significantly when stimulating in VL/VA) or depress (when stimulating in VPM/VPL). Importantly, at 10 Hz, stimulation always drives strong long-lasting feedforward inhibition, which hyperpolarizes cortical cells immediately after each short-latency excitatory response (e.g. see Fig. 1 in Castro-Alamancos and Connors, 1996b). This inhibition of cortical cells may explain the negative BOLD observed in S1 during 10 Hz stimulation (20 sec); under this stimulation regimen, cortical cells will be hyperpolarized most of the time.*

We thank the reviewer for their critical feedback of our writing and data interpretation, and apologize for several key studies being omitted from our Introduction/Discussion that we now include. In particular, we now discuss feedforward inhibition and the possibility that it underlies the observed frequency-dependent silencing of sensory cortex. After careful consideration of existing literature and our own results, we conclude that feedforward thalamocortical inhibition is unlikely to explain the effects reported here. Several lines of evidence point to this conclusion. First, cortical IPSPs driven by feedforward inhibition typically depress after a few repeated stimuli (Cruikshank et al., 2012; Cruikshank et al., 2010). Second, high frequency activity evokes increased levels of interneuron-driven cortical inhibition relative to excitation, whereas low frequency activity does not (Galarreta and Hestrin, 1998). Third, white matter stimulation increases inhibitory synaptic activity in cortex during 40 Hz, but not 10 Hz, stimulation (Contreras and Llinas, 2001). The figure highlighted by the reviewer (Fig. 1 in Castro-Alamancos and Connors, 1996b) is informative in highlighting the frequency-dependent properties of thalamic stimulation, but suggests that 10 Hz stimulation still evokes cortical spikes with every stimulus. We did not observe this type of response, with the average firing rate in cortex being 3.6 ± 3.4 Hz (*n* = 11 neurons, mean ± std) during the 20 s period of 10 Hz stimulation.

To provide additional support that the zona incerta plays a role in the observed cortical inhibition, we performed a new series of experiments in which zona incerta is inhibited using optogenetics during 10 Hz stimulation of central thalamus (Figure 5). We report that targeted inhibition of the zona incerta during 10 Hz central thalamus stimulation reverses the cortical inhibition effect. That is, suppressing zona incerta activity has a net excitatory effect on cortex. These data suggest that zona incerta plays a causal role in contributing to the cortical inhibition reported.

*Second, the idea that the suppression of neural activity observed at 10 Hz would be caused by the zona incerta seems unlikely. There are numerous other possibilities. For example, the very robust drive of feedforward inhibition in neocortex at 10 Hz is more parsimonious. In general, this part of the study is very unconvincing.*

As noted above, we have performed a new series of experiments using optical inhibition of zona incerta to demonstrate this region’s causal role in the frequency-dependent inhibition of cortex. In addition, we have significantly revised the manuscript to discuss alternative hypotheses of the cortical inhibition’s mechanism, including disynaptic thalamocortical inhibition (see Discussion, paragraph four) and disynaptic corticocortical inhibition (see Discussion, paragraph six).

*Third, there are significant methodological issues to consider. Most important is the fact that the authors chose to keep the duty cycle constant across stimulation frequencies. This means that at 10 Hz the light stimulus per cycle lasts 30 ms and at 100 Hz it lasts 3 ms. This is very problematic because the changes that are being attributed to frequency can easily be attributed to pulse duration, which has very significant effects on the responses evoked. Thus, everywhere in the paper where the authors allude to frequency, they are also changing pulse duration, which is a major confound. The only reasonable way to keep the total light duration constant when studying frequency is by increasing the train duration for the lower frequencies and keeping the pulse duration constant.*

As explained in our response to Reviewer #2, we chose to keep the duty cycle constant in these experiments to keep both the total amount of light delivery and pulse train duration constant. Maintaining a constant amount of light delivery is important to mitigate possible confounding factors associated with heating (Christie et al., 2013). Similarly, maintaining constant pulse train duration is necessary to avoid possible differences in neuronal adaptation that would occur when changing the train duration by such a large factor (e.g. an order of magnitude between 10 and 100 Hz).

To provide additional insight regarding this potential confound, we performed additional fMRI experiments keeping both the pulse width and train duration constant. Our results show that the transition from negative to positive BOLD signals in cortex is preserved, as well as the changes in activation volume in cortex and striatum (Figure 2—figure supplement 3). Although these stimulus trains deliver different amounts of total light, the fact that we see these phenomena while varying frequency and keeping either the total amount of light delivery (Figure 2 and Figure 3) or pulse width (Figure 2—figure supplement 3) constant suggests that frequency is the main determinant in driving these effects.

*Fourth, the selectivity of the stimulation, which is based mostly on the placement of the fiber, is a major limitation because the nuclei that are targeted are very small and have complex shapes. It seems inevitable that adjacent structures will also be driven by the light. A Cre-mouse that would limit the expression to the targeted nuclei would be a much better choice in this particular case.*

We agree with the reviewer that the issue of stimulation selectivity is a very important one, and we paid careful attention to it throughout the experimental design process. In particular, we used a large sample size (*n* = 10-16 animals) and ensured that the stimulation location was within a very narrow range across animals using high-resolution in vivo imaging and excluding over half the original cohort (*n* = 47) from analysis. To ensure that stimulation was primarily limited to the targeted nuclei, we also titrated the power level to achieve an activation cone of appropriate scale (Figure 1). Given these measures, the authors are confident in the claim that the effects reported in this manuscript are primarily driven by the intralaminar nuclei of central thalamus. Because there is no genetic marker that differentiates intralaminar nuclei from the rest of thalamus, development of a Cre-mouse line is currently not possible. Nevertheless, we agree that such an experiment would be ideal.

*Fifth, regarding the idea that the thalamus serves to drive cortical activation, previous work has shown that neuromodulation of neural activity in sensory thalamus leads to activation or deactivation in the related sensory cortex (Hirata and Castro-Alamancos, 2010). This work is directly related to the study at hand but was not considered here.*

We sincerely apologize for this oversight. We have added this reference to the revised manuscript.

References cited:

Cruikshank SJ, Urabe H, Nurmikko AV, & Connors BW (2010) Pathway-specific

feedforward circuits between thalamus and neocortex revealed by selective

optical stimulation of axons. Neuron 65(2):230-245.

Galarreta M & Hestrin S (1998) Frequency-dependent synaptic depression and

the balance of excitation and inhibition in the neocortex. Nat Neurosci 1(8):587-

594.

Contreras D & Llinas R (2001) Voltage-sensitive dye imaging of neocortical

spatiotemporal dynamics to afferent activation frequency. J Neurosci

21(24):9403-9413.

[Editors’ note: the author responses to the re-review follow.]

*Several of their original concerns have been satisfactorily addressed in the revised version and the reviewers have acknowledged the improvements made to the manuscript, particularly through the inclusion of the new pulse width and train duration imaging data. Nevertheless, some substantive issues remain that need to be addressed before a final decision can be made (references cited are given below).*

Many of the findings presented in our study are consistent with the anatomical and physiological properties of thalamus and thalamocortical projections that have been documented separately over the last half century. It is notable, however, that our experimental paradigm is unique and fundamentally different compared to those employed in previous studies due to our novel approach of selective excitation of central thalamic relay neurons using optogenetic techniques and the visualization of whole-brain response patterns using fMRI. The several well-characterized thalamocortical properties cited by the reviewer originate from thalamic stimulation in related but distinct experimental preparations (e.g. electrical microstimulation or optogenetic stimulation in reduced preparations), which though clearly directly related, may produce different effects. The precise frequency selectivity of the whole-brain network responses shown here has simply not been demonstrated before (other approaches have been limited to looking at only some pieces of the network at one time). With that said, we agree that many aspects of our findings have interpretations in the context of prior literature. One reviewer relates our findings to the well-known recruiting response, which we agree could possibly co-exist with and/or contribute to the observed responses (see Discussion paragraph five). However, there is a much more clear and compelling comparison to a fourth thalamocortical physiological mode described since the 1940s (in addition to the three already noted by the reviewer) – the production of high voltage spike-wave activity associated with behavioral arrest. This mode is directly supported by our data, as we show that 10 Hz stimulations generate spike-wave EEG and behavioral arrest patterns, and concretely link our stimulation paradigm to the zona incerta (see below). We now more clearly present the evidence to frame and support this interpretation in the revised manuscript.

*1) One of the reviewers argues that your claim that the ZI is the source of cortical inhibition during 10 Hz stimulation is still not well supported. Whilst agreeing that thalamocortical responses evoked in the sensory cortices by stimulating the sensory thalamus at 10 Hz are depressed both in vivo (Castro-Alamancos and Connors, 1996b) and in slices (Cruikshank et al., 2010), the reviewer has pointed out that thalamocortical responses evoked in neocortex by stimulating other thalamic nuclei (midline, intralaminar, and VA/VL) are different in most respects, both in vivo (Castro-Alamancos and Connors, 1996b) and in slices (Cruikshank et al., 2012). Stimulation of these other thalamic nuclei in vivo at 10 Hz produces facilitating (spindle-like) responses in cortex, not depressing responses. In fact, work in slices also supports this. A direct quote from the in vitro paper (Cruikshank et al., 2012) cited by the authors in the rebuttal emphasizes this, "responses during repetitive stimulation were far more sustained for matrix than for core sensory TC pathways".*

We agree with the reviewer’s claim that thalamocortical inhibition mediated by cortical interneurons cannot be ruled out completely. We would like to thank the reviewer for pointing this out and have revised the manuscript accordingly. First, we changed the wording in the Discussion from “A competing hypothesis to explain the suppression of cortex […]” to “An additional mechanism that could contribute to the evoked suppression of cortex […]”. We believe this highlights the possibility that feedforward thalamocortical inhibition may play a role in the observed response, while also acknowledging the novel, causal role of zona incerta discovered in our experiments. Second, we removed our previous argument that thalamocortical inhibition becomes depressed after repeated stimuli, including the citation that referenced stimulation of the VB complex. Furthermore, we make note of Cruikshank et al.’s recent finding that the matrix neurons of central thalamus can generate relatively sustained IPSPs (see Discussion, paragraph four). We conclude that our results are compatible with feedforward thalamocortical inhibition acting in conjunction with the novel zona incerta-modulated inhibition revealed here.

We also added citation (Castro-Alamancos and Connors, 1996b) listed by the reviewer in reference to short-term plasticity of thalamocortical synapses.

*Since the 1940s it has been known that stimulating thalamic nuclei at different frequencies evokes three main types of responses in cortex (Dempsey and Morison): primary responses, augmenting responses and recruiting responses. A more recent account of this literature, including more recent studies done in the rat (Dempsey and Morison worked in cats), is presented in Castro-Alamancos and Connors, 1997 (not cited by the authors, but highly relevant). Primary responses occur when specific thalamic nuclei are stimulated, such as sensory thalamus. Primary responses depress with frequency. Augmenting and recruiting responses occur when VL, VA, VM, midline or intralaminar thalamic nuclei are stimulated. These responses are similar in that they produce enhanced cortical responses at stimulation frequencies between 7-14 Hz. They differ in their cortical depth profile, indicating that they target different layers (as expected from the known projections), and in the size of the primary response they evoke. VL stimulation evokes augmenting responses with a small primary response (small compared to sensory pathways) in motor cortical areas, which is surface positive. In contrast, recruiting responses are mostly surface negative, lacking the surface positive primary response of augmenting responses. The present study did not record field potential responses in cortex, so it is difficult to know if the responses they evoke at 10 Hz in cortex are augmenting or recruiting. But a consistent finding from intracellular recordings is that strong feedforward inhibition is evoked in cortex in vivo by stimulating the thalamus at 10 Hz (Castro-Alamancos and Connors, 1996c; Castro-Alamancos and Connors, 1996a). In fact, the area stimulated by the authors is known to evoke recruiting responses, which should yield feedforward inhibitory responses with weaker short-latency excitation (surface positivity) than augmenting responses. The reviewer concludes that by citing papers based on specific thalamic nuclei, such as core sensory thalamic nuclei (Cruikshank et al., 2010), is not appropriate, because it is well known that the pathways stimulated in this study have different response properties.*

We agree with the reviewer that papers based on stimulation of specific thalamic nuclei should be avoided and removed citation (Cruikshank et al., 2010) from the Discussion. We also include citations (Castro-Alamancos and Connors, 1997) and (Castro-Alamancos and Connors, 1996c) listed above, and discuss the recruiting response and its hypothesized mechanism of hyperpolarization driven by thalamic stimulation. However, we wish to add that the list of three cortical responses described by the reviewer (primary, augmenting, and recruiting) omits another key mode of thalamocortical activation – namely, the generation of spike-wave cortical activity and behavioral arrest that is known in human subjects as the absence seizure and is modeled in various rodent strains. Indeed, the reference provided by the reviewer (Castro-Alamancos and Connors, 1997) also ends with a discussion of this general phenomenon. However, it is not presented as an isolated mode of cortical response, likely because it is a network property and is not well isolated in the experimental contexts in which the other three responses are characterized. We would point out that our experimental framework of whole-brain imaging is uniquely suited to characterize the circuit-level differences of stimulation frequency across the entire brain during sustained, steady-state activation (approximating the typical approach used in deep brain stimulation) rather than the methods used to study primary, augmenting, and recruiting responses (which are generated by shorter duration stimulation paradigms and lack specific behavioral correlates).

Here, we demonstrate that 10 Hz optogenetic stimulation during sleep produces slow spike-wave activity and freezing behavior in the majority of animals, consistent with absence seizure-like responses. We are led to zona incerta directly from these data and existing literature on rodent absence seizures that points to the ZI’s role in modulating spike-wave discharges (see Introduction). We also wish to emphasize that we are not the first group to suggest a key role for incertofugal inhibition in spike-wave activity or to probe this possibility experimentally. For example, one study (Shaw et al., 2013) concluded, “Membrane hyperpolarization due to the intensive inhibitory GABAergic actions is beneficial for maintaining brain rhythms […] Delayed rhythmic incertal activity may help to develop membrane hyperpolarization that builds up sustained high-voltage cortical rhythms through GABAergic incertofugal pathways.” This point has been clarified in the revised manuscript (see Discussion).

*2) The reviewer has also criticized your argument that high frequency activity evokes increased levels of interneuron-driven cortical inhibition relative to excitation, whereas low frequency activity does not, on the grounds that the relevance of the cited paper (Galarreta and Hestrin, 1998) to the issue at hand is doubtful because this study only deals with inhibition driven by cortical stimuli, not thalamic stimuli. You need to look at feedforward inhibition driven by thalamic stimulation, preferably in vivo. The other cited slice paper (Cruikshank et al., 2012), looked at feedforward inhibition driven by thalamic inputs but only marginally (only 1 example panel in Figure 6 and no group data). It is hard to derive any strong conclusions other than to say that inhibition is there, albeit depressed compared to the 1st stimulus.*

We agree with the reviewer and removed this argument (including citation Galarreta and Hestrin, 1998) from the Discussion. Unfortunately, to the best of our knowledge, no study has directly compared the frequency-dependent strength of cortical interneuron-mediated thalamocortical inhibition during intralaminar thalamic stimulation in vivo. Therefore, it is difficult to predict the extent to which this inhibition would vary between our 10, 40, and 100 Hz stimulations. Intracellular recordings would be an appropriate means to further investigate this issue, a discussion point we have added to the revised manuscript. Until these experiments are conducted, our results stand on their own as a novel and exciting demonstration that zona incerta plays a causal role in modulating cortical inhibition induced by central thalamus stimulation. While zona incerta has previously been implicated in modulating spike-wave discharges and associated absence seizures, we were able to link this relationship to specific whole-brain activity patterns and a previously described inhibitory pathway originating from ZI.

*3) The reviewer also points out that white matter stimulation does not mimic thalamic stimulation, let alone central thalamus stimulation, so this should not be used as an argument for the effects of thalamic stimulation on cortical activity.*

We agree that white matter stimulation does not accurately mimic thalamic stimulation and removed this citation from the Discussion. This reference had been included in the previous submission due to the general lack of other studies comparing cortical inhibition between different frequencies of stimulation. We note that the results described in our manuscript help to fill this void.

*4) In response to your comment that, in contrast to Fig. 1 in Castro-Alamancos and Connors, 1996a, you did not observe spikes on every stimulus presentation at 10 Hz, the reviewer points out that this is based on VL stimulation and recording in the motor cortex, which evokes an augmenting response. If in that figure (or Fig. 3 of the same paper) the surface positive field potential component and associated corresponding excitatory drive (which produces the spikes) are eliminated, what is left is a recruiting response, as defined originally by Dempsey and Morison. Similar to spindles, recruiting responses trigger spikes only on some of the waves at 10 Hz. Thus, 3.6 Hz would be consistent with this. It would be helpful if you were to present PSTHs using each stimulus of the 10 Hz train as the marker (Figure 3).*

As requested by the reviewer, we included PSTHs of cortical firing during 10 Hz central thalamus stimulation in the revised manuscript (see Figure 2—figure supplement 4). These figures show that spike events have a non-uniform distribution over time and are more likely to occur after a 6-34 ms latency from the onset of individual light pulses. While such a pattern is expected to occur during disynaptic thalamocortical inhibition (and is therefore compatible with the reviewer’s hypothesis), it should be noted that it does not confirm or implicate the presence of thalamocortical inhibition mediated by cortical interneurons. Instead, this pattern only suggests that the evoked inhibition is not strong enough to completely eliminate the spikes evoked monosynaptically by excitatory thalamocortical afferents. In other words, while the PSTHs are consistent with cortical interneurons driving the observed inhibition (in the sense that they show a stimulus-driven excitatory response), the observed histogram distributions would likely also be present if inhibition was mediated by an external source such as zona incerta.

*5) Your evidence that inhibition of the zona incerta during 10 Hz central thalamus stimulation reverses the cortical inhibition effect is based on AAV delivery of halorhodopsin, which will not discriminate between inhibitory and excitatory neurons. Consequently, light delivery might not only inhibit ZI cells, but other cells too, including adjacent thalamocortical cells. It is therefore possible that the effects observed are due to inhibition of thalamocortical cells. Similarly, could the mCherry fibers observed in cortex actually be thalamocortical fibers? It is also possible that your stimulation could be activating the NRt, which would drive the observed spindles and the recruiting responses in the cortex at 10 Hz, in which case there would be no need for the extrinsic inhibition purportedly provided by ZI. These possibilities should be addressed in the paper. Unless they can be ruled out, conclusions about the role of the zona incerta should be modified accordingly.*

To address this concern, we added a quantitative characterization on the volume of eNpHR-expressing tissue activated by 593 nm light delivery (see Figure 5—figure supplement 1). These calculations indicate that the cone of eNpHR activation predominately falls within the zona incerta. Furthermore, virtually none of the cone includes the thalamus (which already has a low mCherry expression-level compared to ZI), suggesting that the effects observed are not due to inhibition of thalamocortical cells. Notably, the incomplete coverage of zona incerta by the 593 nm cone of activation may explain why inhibition is not fully reversed across all recorded cells in cortex (Figure 5).

A similar argument can be made regarding direct excitation of the reticular nucleus suggested by the reviewer. As shown in Figure 1, the cone of tissue directly excited by 473 nm light is far removed from the reticular nucleus. Thus, it is highly unlikely that such a phenomenon could be driving the observed spindles and inhibition during 10 Hz stimulation.

Regarding the mCherry fibers reported in cortex, it has been known for over two decades that zona incerta projects to sensory cortex (see Lin et al., Science, 1990). Thus, while we agree that the mCherry-expressing axons observed in cortex may not necessarily originate from zona incerta, we also maintain that such specificity is not necessary to support our conclusions on zona incerta-mediated inhibition. As originally stated in the manuscript, these axons are simply consistent with previously reported projections.

*6) In discussing alternative hypotheses of the mechanism of cortical inhibition, including disynaptic thalamocortical inhibition (see Discussion, paragraph four) and disynaptic corticocortical inhibition (see Discussion, paragraph six), you should include relevant literature (e.g. rat in vivo thalamic stimulation), instead of marginally related work (slice studies that used white-matter or cortical stimulation, hippocampal studies, etc.). Care should also be taken to cite the literature accurately. For example, you argue that "central thalamus at different frequencies can lead to distinct behavioral responses" and cite Morison and Dempsey (1942), but Dempsey and Morison did not study behavior. Galarreta and Hestrin (1998) did not study thalamocortical responses (please correct this in the subsection “Central thalamus stimulation frequency controls cortical excitation/inhibition balance”).*

We thank the reviewer for pointing out these errors. We replaced the Dempsey and Morison reference with two studies demonstrating distinct behavioral responses (e.g. arrest/absence seizures) during low-frequency stimulation of intralaminar nuclei in vivo. We also replaced the citation with one that studied thalamocortical responses (Castro-Alamancos and Connors, 1996b), and added studies utilizing in vivo thalamic stimulation in rat to the Discussion (Castro-Alamancos and Connors, 1996a.; Casto-Alamancos and Connors, 1996c). We appreciate how these changes have helped to improve our manuscript.